# Ionic liquid facilitated melting of the metal-organic framework ZIF-8

Vahid Nozari[1], Courtney Calahoo[1], Joshua M. Tuffnell[2], David A. Keen [3], Thomas D. Bennett [2] &
Lothar Wondraczek [1,4✉]

Hybrid glasses from melt-quenched metal-organic frameworks (MOFs) have been emerging
as a new class of materials, which combine the functional properties of crystalline MOFs with
the processability of glasses. However, only a handful of the crystalline MOFs are meltable.
Porosity and metal-linker interaction strength have both been identified as crucial parameters
in the trade-off between thermal decomposition of the organic linker and, more desirably,
melting. For example, the inability of the prototypical zeolitic imidazolate framework (ZIF)
ZIF-8 to melt, is ascribed to the instability of the organic linker upon dissociation from the
metal center. Here, we demonstrate that the incorporation of an ionic liquid (IL) into the
porous interior of ZIF-8 provides a means to reduce its melting temperature to below its
thermal decomposition temperature. Our structural studies show that the prevention of
decomposition, and successful melting, is due to the IL interactions stabilizing the rapidly
dissociating ZIF-8 linkers upon heating. This understanding may act as a general guide for
extending the range of meltable MOF materials and, hence, the chemical and structural
variety of MOF-derived glasses.

[1] Otto Schott Institute of Materials Research, University of Jena, Jena, Germany. [2] Department of Materials Science and Metallurgy, University of Cambridge,
Cambridge, United Kingdom. [3] ISIS Facility, Rutherford Appleton Laboratory, Harwell Campus, Didcot, Oxfordshire, United Kingdom. [4] Center of Energy and
Environmental Chemistry, University of Jena, Jena, Germany. ✉email: lothar.wondraczek@uni-jena.de

Metal-organic frameworks (MOFs) are porous crystalline three-dimensional networks composed of organic linkers coordinated to inorganic metal centers. They are of great interest owing to their structural tunability and potential applications in gas storage and separation, catalysis, drug delivery, and clean water harvesting[1–5]. Research on developing new structures has led to the discovery of over 70,000 MOFs, mostly in the form of polycrystalline powders[6]. The use of such powders in certain applications requires handling and processing into bulk and mechanically stable shapes or geometries. For instance, preparation of pellets is a possible route, however, pellet formation and achievement of the required mechanical stability can be challenging[7]. Alternative routes for the fabrication of bulk, shapeable, and robust architectures with enhanced processability are therefore highly desired, thereby broadening the range of potential MOF applications[8].

Liquid MOFs and melt-quenched MOF glasses from zeolitic imidazolate frameworks (ZIFs) have emerged recently as a new class of materials, offering processable bulk shapes which still retain the advantageous chemical functionality of crystalline MOFs[9]. ZIFs are a subset of MOFs having similar topologies as those which are found in inorganic zeolites (tetrahedral $Zn^{2+}$ are coordinated by imidazolates instead of tetrahedral $SiO_4^{4-}$ and $AlO_4^{5-}$ species bonded via corner-shared oxygens)[10–12]. However, only a handful of ZIFs have been observed to form melt-quenched glasses[13–15]. The limited meltability of crystalline MOFs results from the decomposition temperature ($T_d$) being lower than the melting temperature ($T_m$) of the MOF framework. In the majority of cases, the organic linkers decompose prior to metal-ligand coordination bond breakage and reformation (i.e., melting). This prevents the material from reaching the potential liquid state. Post-processing strategies by which $T_m$ could be reduced to below $T_d$ would enable access to a much more diverse array of MOF glasses. This could open a wide variety of physicochemical properties, and significantly broaden the range of potential applications.

The microscopic mechanism of ZIF melting, the breaking and reformation of Zn–N bonds (referred to as defect formation) has been observed for meltable ZIFs such as ZIF-4 [$Zn(Im)_2$, $Zn(C_3H_3N_2)_2$], ZIF-zni [$Zn(Im)_2$, $Zn(C_3H_3N_2)_2$], and ZIF-62 [$Zn(Im)_{2-x}(bIm)_x$, $Zn(C_3H_3N_2)_{2-x}(C_7H_5N_2)_x$ for $0 < x < 0.35$][13,14,16]. This mechanism occurs via rapid dissociation and replacement of an initially coordinated linker with a neighboring linker[9]. Melting requires that the vibrational displacement of atoms in the crystal

structure reaches a characteristic level (instability)[13], which is achieved by heating[17]. However, in the more open networks such as ZIF-8 [$Zn(mIm)_2$, $Zn(C_4H_5N_2)_2$], the (calculated) temperature at which Lindemann's ratio reaches the threshold for melting by far exceeds those of ZIF-4 (1200–1500 K) and ZIF-zni (1500–1750 K): the higher energy barrier for linker mobility in ZIF-8 precludes framework melting[18].

ZIF-8, a commercially available ZIF with sodalite topology, has been investigated extensively in the literature for a wide range of applications such as microelectronics[19], catalysis[20], drug delivery[21], and gas separation[22]. Theoretical studies on ZIF-8 melting revealed that the bond cleavage activation enthalpy and entropy of ZIF-8 exhibit a significant difference (43% for enthalpy and 60% for entropy) between Zn–N and Zn–Im coordination (where Im is the center of mass of the imidazolate linker). However, for other ZIFs such as ZIF-4 and ZIF-zni variations of activation enthalpy and entropy in Zn–N and Zn–Im are less than 3%. This observation showed that Zn–N bonding strength is not the only parameter determining meltability. The specific behavior of ZIF-8 was further confirmed in simulation studies which found Zn to retain fourfold coordination up to 1250 K (in silico). The extent of interionic interactions, i.e., interactions between metal cations and organic anions, is, therefore, a crucial factor for melting; weaker interionic interactions facilitate melting[18].

The energy of defect formation was found to be similar for ZIF-8, ZIF-4, and ZIF-zni, i.e., 71, 56, and 67 kJ mol$^{-1}$, respectively[18]. The striking difference between these three ZIFs is in their surface area (or porosity); ZIF-8 has a dramatically higher porosity as compared to the other two ZIFs (~1200 vs. 400 and 4 m$^2$g$^{-1}$)[23–25]. This difference is even more evident (see Fig. 1a) when comparing the pore diameter $d_p$ of ZIF-8 (11.6 Å) with those of ZIF-4 (2.1 Å) and ZIF-62 (1.3 Å), Fig. 1a[26], suggesting that porosity is a key factor determining meltability.

There is thus a major constraint which prevents the melting of ZIF-8. Specifically, the relatively high porosity of the framework, which is linked to the absence of charge stabilization of the newly dissociated linker[18]. Hence, melting should occur where the high free energy (stemming from the highly porous nature of ZIF-8) and interionic interactions between the metal cation and organic anion are both diminished. Following this hypothesis, we incorporated an ionic liquid (IL), 1-ethyl-3-methylimidazolium bis(-trifluoromethanesulfonyl)imide, [EMIM][TFSI] into ZIF-8 pores aiming to decrease the $T_m$ of ZIF-8 to below its $T_d$, and reaching

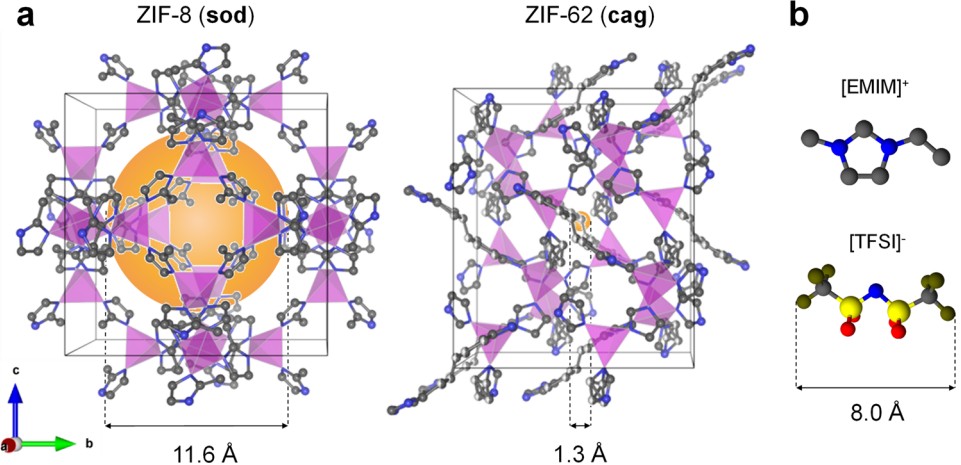

**Fig. 1 Crystal structure of ZIF-8 and ZIF-62 and molecular structure of [EMIM][TFSI] used in IL@ZIF-8 composite. a** Crystal structures and pore diameters of ZIF-8 and ZIF-62. The orange spheres in the crystal structures are drawn to mark the free space inside the cages[26]. Crystallographic data of ZIF-8 and ZIF-62 are taken from literature[24, 25]. **b** Molecular structure of [EMIM][TFSI]. Color codes: Zn – purple tetrahedra, N – blue, C – gray, S – yellow, O – red, F – olive. H – omitted for clarity.

the liquid state as a result of interactions between the IL and ZIF-8 at elevated temperatures.

Interactions between different ILs and MOFs have been extensively investigated experimentally and computationally at low temperatures. It has previously been shown that the interactions between IL molecules and the MOF structure are crucial in creating new functional sites favorable for adsorption, catalysis, and ion conduction[27]. For example, a simulation study investigated IRMOF-1 supported IL membranes for $CO_2$ capture. Four different ILs with fixed cation and different anions were used to demonstrate that the anion of the IL plays an important role in the extent of interactions between IL and MOF[28].

Combined density functional theory (DFT) calculations and experimental vibrational spectroscopy have also been used to probe the molecular interactions between a Cu-based MOF, copper benzene-1-3-5-tricarboxylate (CuBTC), and an IL, 1-ethyl-3-methylimidazolium ethyl sulfate ([EMIM][EtSO$_4$]). Here, the results showed that interactions between the IL anion and $Cu^{2+}$ ions caused the transfer and redistribution of electron density over the metal sites. A corresponding red-shift was observed in the experimental vibrational spectra in IR bands associated with Cu–O bonding. It was concluded that intermolecular interactions between the linker molecules and Cu weaken upon simultaneous interaction with IL ions[29]. Weakening metal-ligand bonding was further shown via incorporating seven different imidazolium-based ILs into CuBTC pores. It was shown that when the interionic interaction within the IL was higher, the IL was interacting strongly with the structure and Cu–O bonding became weaker, resulting in lower thermal stability of the IL@MOF composites[30].

Here, we use the synergistic concepts of (i) an adjustment of metal-linker bond strength and (ii) a greater extent of energetic stabilization of a newly dissociated linker, to investigate the IL mediated melting of a prototypical porous framework, ZIF-8.

## Results

[EMIM][TFSI] was chosen since it is a hydrophobic IL, enabling incorporation into the hydrophobic pores of ZIF-8[31]. It has a very high $T_d$ (~440 °C) compared to other imidazolium-based ILs[31,32]. 35 wt% of [EMIM][TFSI] were loaded into ZIF-8 using a wet impregnation technique (see Methods section for further details). The resultant composite is herein referred to as IL@ZIF-8 (Fig. 1). The IL loading was adjusted in such a way as to obtain a powder sample without the presence of excess liquid.

The IL@ZIF-8 composite was characterized using X-ray diffraction (XRD) (Fig. 2a), scanning electron microscopy (SEM) (Supplementary Fig. 1), and Fourier transform infrared spectroscopy (FTIR) (Fig. 2b). XRD and SEM results confirmed that IL incorporation did not damage the crystal structure and morphology of ZIF-8. FTIR measurements were carried out in order to examine the incorporation of IL into ZIF-8; they show that all IL IR features are present in the composite sample. These results are in agreement with previous studies on crystalline IL@MOF composites[33–36].

To study the bonding interactions between the IL and ZIF-8 at high temperature, thermogravimetric analyses (TGA) coupled with differential scanning calorimetry (DSC), TGA-DSC, were done on ZIF-8, the IL, and the IL@ZIF-8 composite (Fig. 2c and Supplementary Fig. 2). No phase transitions were observed in pristine ZIF-8 and the bulk IL before the start of decomposition at around 550 and 440 °C, respectively. A small endothermic peak at 381 °C was noted in the IL@ZIF-8 composite, very close to the decomposition temperature (~412 °C). To properly assign this feature to melting, IL@ZIF-8 was heated at 387 and 390 °C under nitrogen (slightly above $T_m$, defined as the offset temperature of the melting peak) for 30 and 40 min, defined as LT (low

temperature) and HT (high temperature) conditions, respectively. After heating, samples were cooled down to room temperature at a rate of 50 °C·min$^{-1}$. The obtained samples are henceforth referred to as a$_g$(IL@ZIF-8-LT) and a$_g$(IL@ZIF-8-HT).

Throughout these experiments, the heating temperature and time were selected in such a way that fully and partially amorphous samples could be acquired for HT and LT conditions respectively, as demonstrated by XRD analysis (see Fig. 2a). The XRD pattern of a$_g$(IL@ZIF-8-HT) contains the broad diffuse scattering characteristic of glass (with a small unidentified Bragg peak at 11.6°). A pure sample of ZIF-8 subjected to the same (HT) treatment retained its crystallinity. However, a$_g$(IL@ZIF-8-LT) contained weak diffuse scattering, alongside Bragg peaks reminiscent of the starting crystalline phase. This suggests that treatment temperature and time are important parameters in the formation of IL@ZIF-8 glasses and crystal-glass composite samples, as indicated in Supplementary Fig. 3 for a range of tested synthesis conditions. A DSC upscan (Fig. 2c) performed on a$_g$(IL@ZIF-8-HT) revealed a glass transition temperature ($T_g$) of 322 °C (595 K), confirming the glassy nature of this sample. With the melting temperature of ~381 °C (654 K), this results in a nominal value of $T_g/T_m$ of ~0.91, which even surpasses the ultrahigh glass-forming ability of 0.84 which was reported for melts of ZIF-62[37]. To more clearly observe the glass transition, referenced DSC runs were conducted in order to extract the isobaric heat capacity $C_p$ (denoted $C_p$ scans). Cyclic $C_p$ scans were performed on a$_g$(IL@ZIF-8-HT) samples at various heating and cooling rates (see Methods section for further details). Figure 2d shows a cyclic $C_p$ scan for a$_g$(IL@ZIF-8-HT) using 20 °C·min$^{-1}$ as heating and cooling rate. A pronounced glass–liquid transition is detected at ~328 °C, with a configurational heat capacity $\Delta C_p$ of ~0.11 J·g$^{-1}$·K$^{-1}$ (Fig. 2d). The magnitude of the jump in $C_p$ is comparable to the values observed for other ZIF glasses such as ZIF-4, 0.11 and 0.16 J·g$^{-1}$·K$^{-1}$ for LDA and HDA phases, and 0.19 J·g$^{-1}$·K$^{-1}$ for ZIF-62[37,38]. The minor up-shift in the glass transition as compared to the DSC scan (Fig. 2c) is attributed to the higher scanning rate (20 °C·min$^{-1}$ vs. 5 °C·min$^{-1}$). Although the second upscan exhibits a similar $\Delta C_p$ to the first, a slight delay is found in the glass transition. We attribute this observation to the continuing interaction between the glass phase and residual IL, simultaneously overlapping with ongoing IL decomposition during extended exposure of the glass to high temperatures in the first upscan and subsequent down scan (which is why we limited the scanning range to the upper limit of 360 °C). Additional cycle $C_p$ scans with different cooling/heating rates are provided in Supplementary Fig. 4. For a$_g$(IL@ZIF-8-LT), shown in Supplementary Fig. 5, the glass−liquid transition is not as clearly visible as in the HT sample, which is because crystalline ZIF-8 (together with residual IL) remains the primary phase in this case, and neither ZIF-8 nor the IL exhibit a DSC feature in this temperature range (Fig. 2c). However, in the second upscan, a weak glass transition is detected also for the LT sample, which is in line with our interpretation of progressive reactions during DSC scanning.

Confocal microscopy images displayed in Fig. 3 show clear evidence of macroscopic flow as a result of melting, as well as direct light transmittance and smooth glass-like surfaces. Macroscopic flow and melting of the IL@ZIF-8 heated from room temperature to 390 °C were observed in situ using a laser scanning microscope (LSM), whereas no morphological changes were seen when heating the parent ZIF-8 to the same temperature (observations are provided as Supplementary Movies 1 and 2 for ZIF-8 and IL@ZIF-8, respectively. All these observations show that ZIF-8, a non-meltable MOF, transforms into a meltable, glass-forming composite through the incorporation of an IL into its pores.

To uncover the microscopic mechanism which facilitates the melting of the IL@ZIF-8 composite, FTIR, TGA,

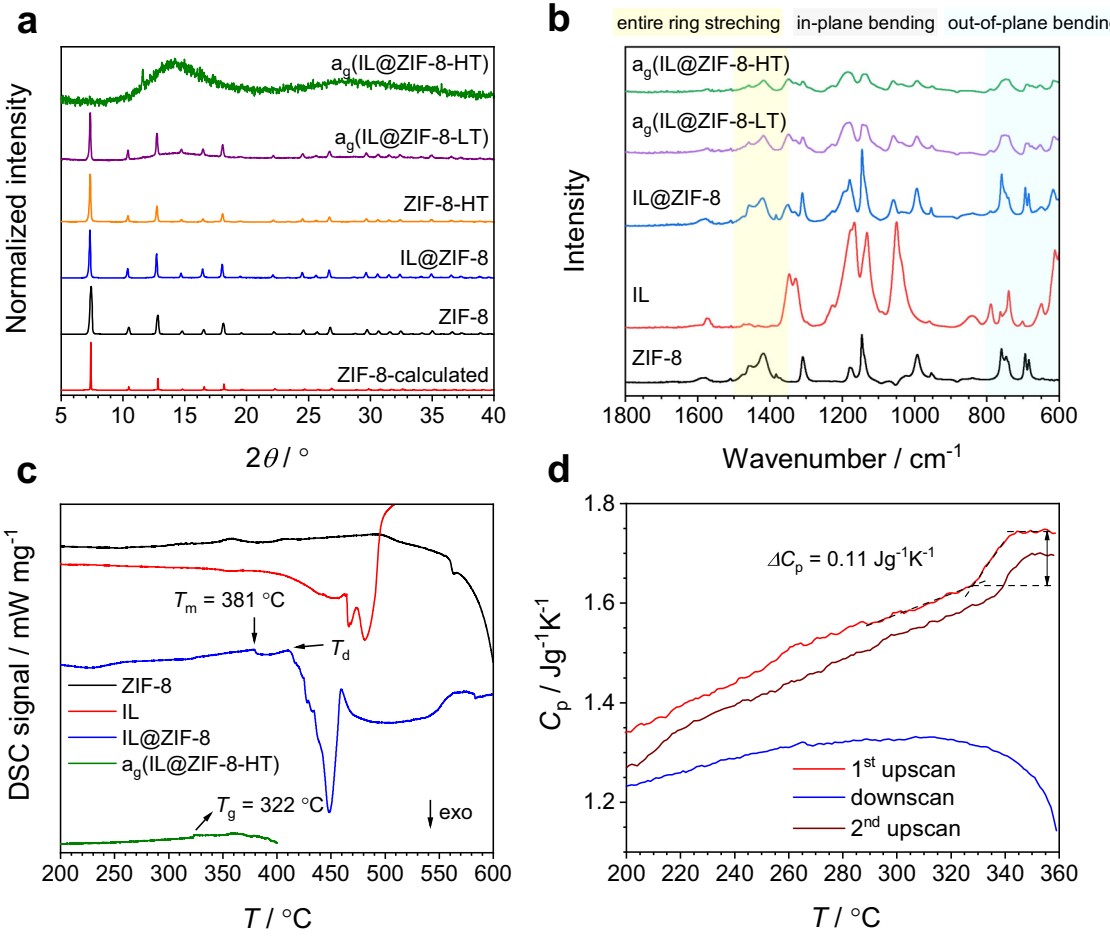

**Fig. 2 Structural characterization, enthalpic responses, and cyclic $C_p$ scan. a** XRD patterns of ZIF-8, IL@ZIF-8 crystalline composite, $a_g$(IL@ZIF-8-LT), and $a_g$(IL@ZIF-8-HT) samples. Crystallographic data were taken from literature[25]. **b** FTIR spectra obtained for ZIF-8, IL, crystalline IL@ZIF-8 composite, $a_g$(IL@ZIF-8-LT), and $a_g$(IL@ZIF-8-HT). **c** DSC scans of ZIF-8, IL, IL@ZIF-8, and $a_g$(IL@ZIF-8-HT) samples with a heating rate of 5 °C·min⁻¹. $T_m$ and $T_d$ are indicated as offset temperatures of the melting peak and onset temperature of decomposition of IL@ZIF-8, respectively. $T_g$ is defined as the onset temperature of the glass transition feature of $a_g$(IL@ZIF-8-HT). **d** Cyclic $C_p$ scan of $a_g$(IL@ZIF-8-HT) with heating and cooling rates of 20 °C·min⁻¹.

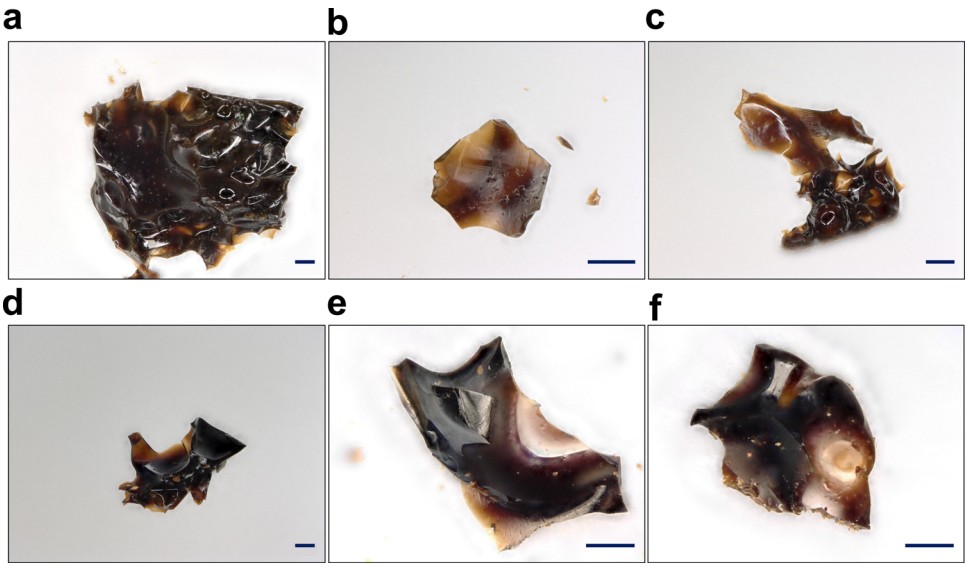

**Fig. 3 Confocal microscopy images showing evidence of melting and glass formation. a–c** $a_g$(IL@ZIF-8-LT). **d–f** $a_g$(IL@ZIF-8-HT). Scale bars are 100 μm.

thermogravimetric analysis coupled with mass spectrometry (TG–MS), $^1$H NMR, $^{13}$C NMR, and total scattering measurements were performed on ZIF-8, IL, IL@ZIF-8, a$_g$(IL@ZIF-8-LT), and a$_g$(IL@ZIF-8-HT) samples. As expected, IR bands in the glassy composites are broader compared to crystalline ZIF-8 and the IL@ZIF-8 composite (see Fig. 2b). Further analysis of deconvoluted spectra in Supplementary Figs. 6 and 7, corresponding to 600–800 cm$^{-1}$ (out of plane bending of imidazole ring) and 800–1700 cm$^{-1}$ (in-plane bending and entire ring stretching of imidazole ring) regions, revealed that the IR bands belonging to ZIF-8 are shifted in IL@ZIF-8, a$_g$(IL@ZIF-8-LT), and a$_g$(IL@ZIF-8-HT) samples. The shifts are summarized in Supplementary Tables 1 and 2. They reflect a clear difference in the interactions between ZIF-8 and IL in crystalline IL@ZIF-8, the melt-quenched glass, a$_g$(IL@ZIF-8-HT), and the crystal-glass composite a$_g$(IL@ZIF-8-LT). The shifts are significantly larger in a$_g$(IL@ZIF-8-LT) and a$_g$(IL@ZIF-8-HT) compared to crystalline IL@ZIF-8, representing stronger electrostatic interactions between the metal center and organic linker of ZIF-8 and the anion and cation of the IL component in these samples, respectively. The shifts show that most of the IR bands of ZIF-8 are shifted to lower frequencies (redshifted), indicating that intramolecular bonding within the 2-methylimidazolate ring of ZIF-8 becomes weaker as a result of intense interaction with IL ions, which only occurs at higher temperatures[29,39]. The resulting interaction becomes stronger when temperature and heating time increase, as evidenced by larger redshifts in IR features of a$_g$(IL@ZIF-8-HT) compared to a$_g$(IL@ZIF-8-LT).

The thermal stability of the bulk IL, pristine ZIF-8, and IL@ZIF-8 composites was examined with the same thermal treatment used to melt the a$_g$(IL@ZIF-8-LT) and a$_g$(IL@ZIF-8-HT) samples. TGA measurements are presented in Supplementary Fig. 8, demonstrating the differences between IL vs. IL@ZIF-8 weight losses. Corresponding quantitative data are provided in Supplementary Table 3. Consistent with the XRD result obtained for ZIF-8-HT, pristine ZIF-8 shows almost no mass loss (1.0 and 0.9% for LT and HT conditions, respectively), while the bulk IL loses 17.5 and 50.0% of its initial mass when heated to LT and HT conditions, respectively, attributed to the decomposition of IL that happens mostly in the isothermal heating step (see Supplementary Fig. 8). The thermal stability of ILs has been studied in dynamic and isothermal TGA experiments previously[40]. ILs mostly decompose at lower temperatures when heated isothermally as compared to the onset decomposition temperature in dynamic heating conditions[32,40–42]. In the present case, the IL@ZIF-8 composite showed 20.7 and 34.4% weight loss for LT and HT heating conditions, respectively.

To understand whether the decomposed species are from IL or ZIF-8 in IL@ZIF-8, we probed the possible decomposition products by conducting TG–MS analysis. TG–MS experiments were performed on bulk IL and IL@ZIF-8 composites with LT and HT heating conditions. As for the decomposition of bulk [EMIM][TFSI], it was previously found that elimination and nucleophilic substitution are major mechanisms of decomposition[43]. At high temperatures (over 350 °C), decomposition of the anion to more nucleophilic groups such as NH$_2$ and F, and the subsequent attack of cation methyl and ethyl groups resulted in the occurrence of different decomposition products in isothermal and scanning TG-MS experiments (in order to minimize the progressive effect of such reactions on the cyclic $C_p$ scans performed on quenched IL@ZIF-8 glasses, Fig. 2d, the upper limit of these scans was set at 360 °C; on the other hand, a further reduction of the scanning range would compromise accurate assignment of the glass transition region). According to the results shown in Supplementary Fig. 9, almost all mass to charge ratios, $m/z$, coming from IL@ZIF-8 match the masses detected from the bulk IL at LT

and HT conditions. Assignment of $m/z$ values to decomposition products has been reported previously[41,43]. Moreover, TG–MS shows that detection of masses occurs in the isothermal segments of LT and HT heating conditions. This agrees with mass losses observed in TGA experiments (see Supplementary Fig. 8).

Digested liquid $^1$H NMR was also performed on ZIF-8, IL, IL@ZIF-8, a$_g$(IL@ZIF-8-LT), and a$_g$(IL@ZIF-8-HT) samples to ascertain the stability of the ZIF-8 linker and of the IL; the spectra are discussed in the Supplementary Information. The results suggest large-scale decomposition of the IL and some linker decomposition within the glass, as also indicated by the darkened color in the optical images (Fig. 3).

The top of Fig. 4 compares the $^1$H–$^{13}$C cross-polarization (CP) NMR spectra for ZIF-8, IL@ZIF-8, a$_g$(IL@ZIF-8-LT), and a$_g$(IL@ZIF-8-HT). CP experiments result in much higher signal/noise (S/N) than single-pulse experiments, but also only allow observation of solid-like carbons ($^1$H $T_1$ times must be longer than the time needed for $^1$H–X polarization transfer)[44]. There are three main carbon peaks highlighted with blue boxes from the mIm linker of ZIF-8: CH$_3$ (C$_1$), CH (C$_{4,5}$), and C (C$_2$) at 14.26, 124.67, and 151.66 ppm; the peaks from pure ZIF-8 are sharp with widths of 0.2–0.3 ppm indicating crystallinity. In comparison, a broad shoulder emerges in a$_g$(IL@ZIF-8-LT) and almost all sharp peaks are absent in a$_g$(IL@ZIF-8-HT), agreeing with the XRD results displayed in Fig. 2a. Although these NMR experiments only probe short-range interactions, the broadness and sharpness of the peaks are clear indications of the degree of crystallinity, allowing for assignment of the peaks to amorphous and crystalline features in Fig. 4. The variety of electronic environments found in broad NMR peaks is assumed to be from varying bond angles and bond lengths, and strongly indicates a system without long-range order. This loss of crystallinity is confirmed for the a$_g$(IL@ZIF-8-LT), and a$_g$(IL@ZIF-8-HT) by both XRD and pair distribution function (PDF) measurements, as well as by single-pulse $^{13}$C NMR (Supplementary Figs. 16–18).

For a more thorough discussion of the $^1$H–$^{13}$C spectra, we turn to the fits of the C (C$_2$), CH (C$_{4,5}$), and CH$_3$ (C$_1$) peaks, respectively, in Fig. 4b. Upon IL addition, the three main carbon peaks remain mostly unchanged, yet, a substantially shifted second peak emerges downfield (higher ppm) of the CH$_3$ (C$_1$) and CH (C$_{4,5}$) peaks. Since the intensity of CP peaks in liquids is very low (as can be seen for pure EMIM peaks in IL@ZIF-8 in Fig. 4a and in Supplementary Fig. 22), these new peaks in the IL@ZIF-8 in Fig. 4 correspond to ZIF-8 interacting with the IL. Moreover, Supplementary Fig. 24 confirms the appearance of a new peak at ~15.4 ppm in IL@ZIF-8 that does not exist in either pure IL or ZIF-8. Finally, IL-ZIF-8 interactions are further corroborated by the different chemical shifts of IL vs. IL@ZIF-8 in Supplementary Fig. 23.

For the HT condition, a$_g$(IL@ZIF-8-HT), there are at least three types of carbon peaks (Fig. 4b). The first type of peak, very slight retention of the sharp ZIF-8 and IL-associated sharp ZIF-8 peaks, is in simple agreement with the XRD results. We note that the LT condition shows the same trends (Supplementary Fig. 21). Like the IL-associated carbon peaks, the broad peaks (Fig. 4b) are found downfield of the sharp peaks, indicative of interactions with IL and/or strain of ZIF-8 linkers. The third type of peak is located upfield (lower ppm), identified as free ZIF-8 linkers which are unbonded to Zn$^{2+}$[45]. Overall, the spectral regions representing amorphous features and the free linker are very wide and sometimes contain more than one clear peak; these large ppm ranges represent the many different types of chemical environments and bonding which exist after heating. Returning to more quantitative single-pulse $^{13}$C NMR, in Supplementary Fig. 25, we find that for the C$_2$ carbon in the LT sample roughly 21, 58, and 21% can be identified as crystalline, amorphous, and free linker, respectively.

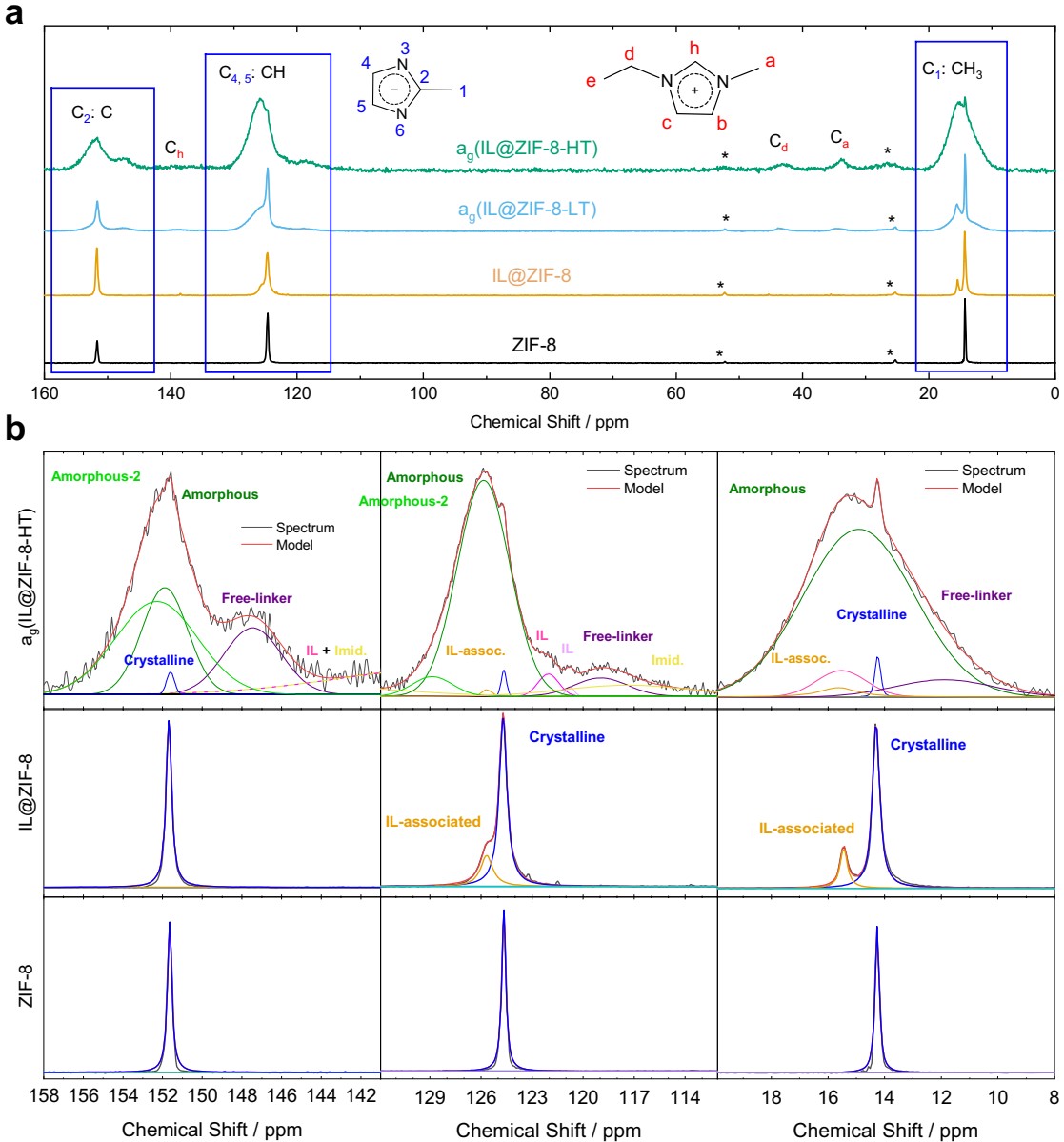

**Fig. 4 ¹H-¹³C CP NMR analysis of mIm carbons in ZIF-8. a** ¹H-¹³C CP NMR of ZIF-8, IL@ZIF-8, a$_g$(IL@ZIF-8-LT), and a$_g$(IL@ZIF-8-HT). Spinning sidebands are marked with asterisks. **b** Insets of fits for C$_2$:C, C$_{4,5}$:CH, and C$_1$: CH$_3$ (methyl) carbons. For the fits of a$_g$(IL@ZIF-8-LT) please see Supplementary Fig. 21.

Simultaneously, in Fig. 5a and Supplementary Fig. 23, we also find a large increase in S/N for the IL peaks in a$_g$(IL@ZIF-8-LT) and a$_g$(IL@ZIF-8-HT), indicating that much more of the IL is now immobilized and behaving like a solid. This result is unsurprising in consideration of the collapse of the pores as evidenced by the dramatic differences in the out-of-plane bending and C=N stretches in the IR after heating (see Supplementary Tables 1 and 2). The most substantial changes in chemical shifts, i.e., electronic environments, are observed for the C$_d$: CH$_2$ and C$_a$: CH$_3$ carbons in EMIM (see Supplementary Fig. 23). In the heat-treated samples, we further observe the formation of imidazole, highlighted in gold, confirming the loss of ethyl and methyl from EMIM observed in the ¹H NMR results. Finally, for the a$_g$(IL@ZIF-8-HT) sample, the signals from C$_h$ in the IL cation and C$_1$ in imidazole are very broad. This is in line with our ¹H NMR and literature data which reveal this hydrogen to be the most reactive and likely to interact with other molecules in the material[46,47].

The ¹H–¹⁵N CP NMR spectra for ZIF-8, IL@ZIF-8, and a$_g$(IL@ZIF-8-LT) are provided in Fig. 5b, perfectly matching the trends of ¹H–¹³C CP NMR, Fig. 4a. Upon heating, a broad shoulder appears downfield along with a broad peak upfield identified again as free-linker. The free-linker peak position represents a much different environment than that of the original ZIF-8 linker. This is expected, given that the nitrogen of the ZIF-8 linker bonds directly to the Zn²⁺ metal center: the shift direction suggests the formation of Zn–H bonding as the imidazole peak is found at −171.6 ppm in DMSO[48]. This general increase in shielding, i.e., electron density, on the nitrogen of the mIm applies for the carbon atoms of the ZIF-8 linker as well, as all the assigned free-linker peaks have lower chemical shifts than their corresponding intact framework peaks. Comparing to ¹H-¹⁵N CP NMR of a$_g$(IL@ZIF-8-LT), the sharp peak which would be evidence for a crystalline phase is not detected in ¹H-¹⁵N CP NMR of a$_g$(IL@ZIF-8-HT), shown in Supplementary Fig. 26, which is in agreement with XRD data in Fig. 2a.

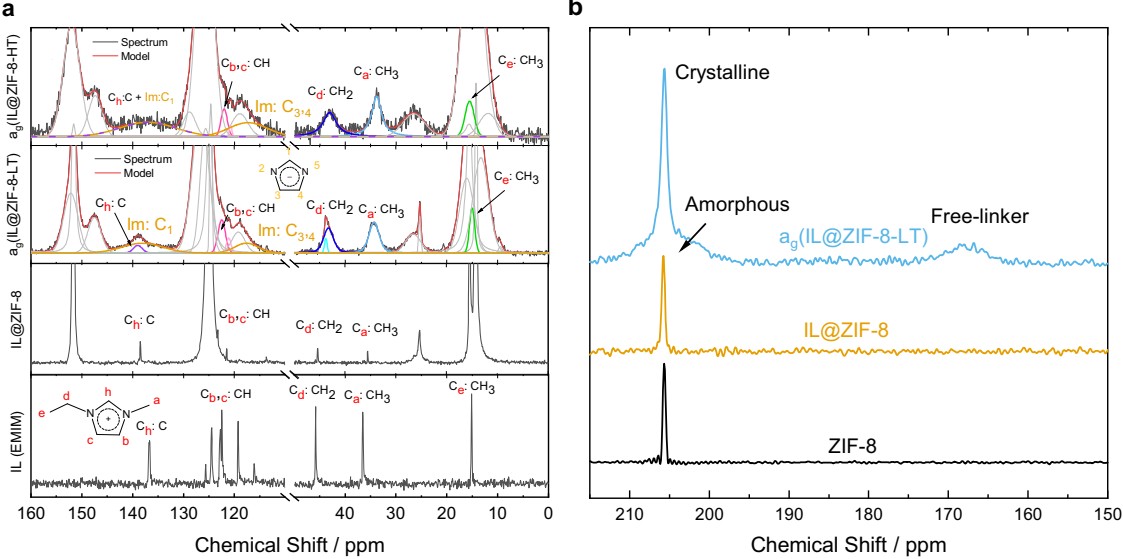

**Fig. 5 NMR analysis of EMIM carbons and mIm nitrogen of the ZIF-8 linker. a** EMIM NMR Signatures Comparison. Decoupled $^{13}$C SSNMR of IL, $^1$H-$^{13}$C CP SSNMR of IL@ZIF-8, $a_g$(IL@ZIF-8-LT), and $a_g$(IL@ZIF-8-HT). Pronounced broadening and shifting of the IL carbon peaks in $a_g$(IL@ZIF-8-LT) and $a_g$(IL@ZIF-8-HT) confirms immobilization of the IL and collapse of ZIF-8 pores at elevated temperatures. For clarity, IL cation peaks are highlighted, while the ZIF-8 peaks are grayed out in the heat-treated samples. In the $a_g$(IL@ZIF-8-LT) and $a_g$(IL@ZIF-8-HT) spectra appearance of Im cations can be observed in gold. **b** $^1$H-$^{15}$N CP NMR Comparison of ZIF-8, IL@ZIF-8, and $a_g$(IL@ZIF-8-LT).

According to the literature, strong interionic interactions occur between [EMIM] and [TFSI] within the IL[49]. Moreover, it has been shown that IL ([BMIM][TFSI]) interacts with ZIF-8 via the N or S atoms of the IL anion and Zn sites in ZIF-8[35]. Although ILs are composed of charged ions, they also are molecules with substantial intermolecular hydrogen bonding[50]. For example, the S = O group in trifluoromethanesulfonic acid has been found to form a hydrogen bond with a CH carbon in an imidazole ring[51,52]. Surprisingly to most chemists, several studies show the protons attached to sp$^3$ carbons (methyl and ethyl groups) of EMIM forming H-bonds with halogens and nitrogen[33,50]. We are unsure of the strength of the interaction between $CH_3$ and an H-acceptor such as S = O, but our $^1$H–$^{13}$C CP NMR agrees: the interactions between ZIF-8 linker hydrogens and IL ions explain the reason that only the ZIF-8 linker CH ($C_{4,5}$) and $CH_3$ ($C_1$) carbons are affected by the incorporation of IL, while the ZIF-8 linker $C_2$ or lone C is mostly unaffected by IL incorporation. Furthermore, hydrogen bonding results in positive shifts (deshielding) of carbon peaks in conjugated ring systems, just like our emerging peaks in IL@ZIF-8 in Fig. 4[53]. Thus, we believe the IL anion to interact with the ZIF-8 mIm linkers via H bonding as shown in Fig. 6a.

We know from DSC that melting of the composite occurs well before the substantial mass loss (381 °C vs. 410 °C). Given the evidence of amorphization and formation of free linkers from the solid-state NMR, alongside the proposed mechanism of melting in ZIFs[9], we suggest that upon dissociation from the Zn$^{2+}$ metal centers, the 2-methylimidazolate linkers are stabilized by electrostatic interactions with the IL ions (or partially decomposed IL molecules), which leads to a stable liquid (see Fig. 6b, c).

After incorporation into ZIF-8 and subsequent heating at LT and HT conditions, mass spectroscopy and solution $^1$H NMR tell us that much of the IL anion is lost and that the IL cation loses some of its methyl/ethyl groups upon heating. The leaving groups and percentages of each, based on the area under the TG–MS curves, are shown in Fig. 7a. The mass loss curves were also used to calculate the elemental composition of the final LT and HT

samples, in Fig. 7b, c, respectively. The general trends are that zinc remains, while fluorine, oxygen, and sulfur content decrease substantially in the LT sample and are absent entirely in the HT sample. Commensurately, the carbon and nitrogen concentrations also increase, however, they do so in an unexpected fashion: the increase in nitrogen content is more than double that of the carbon content. Indeed, the many possible decomposition products from the IL make it challenging to discern the exact structure of the resulting glass after further heating. Nevertheless, one possible structure for the LT sample is depicted in Fig. 7b, where some of the original IL cation and anion can be found, as some F, S, and O remains, but where there are also new molecules, such as imines (ketimines, aldimines, sulfinyl imines, and fluoro-substituted amines), which contain the elements known to remain, and are liquids at RT and fairly stable in the absence of water (expected from our Brunauer–Emmet–Teller (BET) measurements in Supplementary Fig. 32 and Supplementary Table 9), especially in the case of hexafluoroacetone, $(CF_3)_2CNH$.

In the HT sample, only Zn, C, N, and H remain in the material, yet again the increase in N content is notably high. Consequently, we believe only imines (ketimines and aldimines) without fluorine substitution remain, which are stabilized by interactions with the Zn$^{2+}$ sites, as has been shown with silver in the literature[54]. In fact, if the R′ group is not H, these molecules are specifically known as Schiff bases when they act as negatively charged ligands to form metal complexes[55]. For example, such an R′ group could be resulting from one of the ethyl/methyl groups that are lost from the EMIM cation, as shown by both the solution $^1$H NMR and $^{13}$C SSNMR. Although the exact composition of the partially decomposed IL and ZIF-8 mixture is hard to determine, the mass loss from the ZIF-8 linkers is low and the net result is the formation of a glass.

We investigated the removal of IL residues from the $a_g$(IL@ZIF-8-HT) and $a_g$(IL@ZIF-8-LT) samples by conducting post-process washing experiments in acetone. Details are provided in the Supplementary Information, Supplementary Fig. 29–32, and Supplementary Table 9 for the obtained results. In short, washing

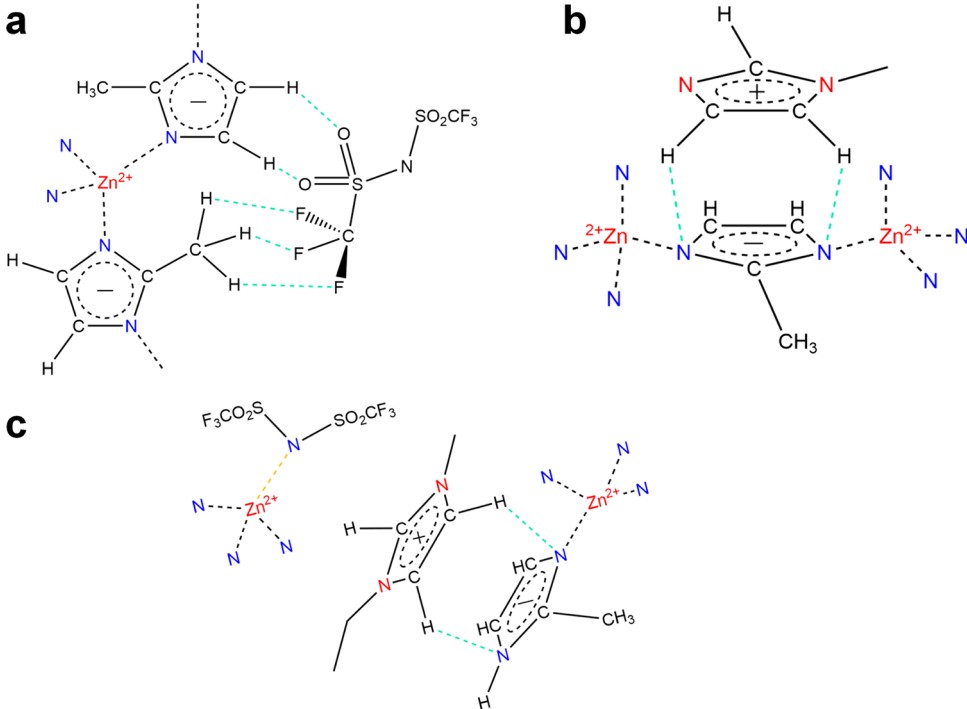

**Fig. 6 2D schematic of possible interactions between IL and ZIF-8. a** Interactions with the IL anion; **b** Interactions with the IL cation. **c** Melting/amorphization of IL@ZIF-8 at 381 °C. Orange and green bonds represent $Zn^{2+}-N^-$ bonding and H bonding between IL and ZIF-8, respectively. For a schematic view of the whole pore, see Supplementary Fig. 27.

results in a significant reduction of the brownish tint (which we have assigned to the decomposition products) and notably enhanced gas adsorption performance by partially removing the unreacted or decomposed compounds from the glass matrix. TGA-DSC scanning of post-washed $a_g$(IL@ZIF-8-HT), Supplementary Fig. 31, confirmed that the glassy nature of the sample is preserved during washing. Gas adsorption experiments using $N_2$ or $CO_2$ (Supplementary Fig. 32) reveal a fourfold increase of the glass' total porosity after washing; in particular, the washed $a_g$(IL@ZIF-8-LT) sample which contains a major fraction of crystalline ZIF-8 (being only partially amorphized) exhibited improved $CO_2$ uptake compared to the adsorption capacity observed for other MOF glasses such as $a_g$ZIF-62 and $a_g$[(ZIF-8)$_{0.2}$(ZIF-62)$_{0.8}$][56,57]. While outside of the scope of the present study, future exploration of LT-type composite materials containing crystalline ZIF-8 in a partially melted IL@ZIF-8 matrix phase may be beneficial towards the application of these materials, for example, in gas adsorption and separation.

To further investigate, and compare, the structures of the crystalline and glassy composites, room temperature synchrotron X-ray total scattering experiments were conducted on the IL, ZIF-8, IL@ZIF-8, $a_g$(IL@ZIF-8-LT), and $a_g$(IL@ZIF-8-HT) samples. For comparison with the heat-treated samples, $a_g$(IL@ZIF-8-LT) and $a_g$(IL@ZIF-8-HT), a sample of ZIF-8 was amorphized (see Supplementary Fig. 33) via ball milling for 30 min at 30 Hz in a shaker-type grinding mill—as detailed in the Methods section. This ball-milled sample is herein referred to as $a_m$ZIF-8, in accordance with prior nomenclature[15].

The structure factors, $S(Q)$ in Supplementary Fig. 34a show Bragg peaks for the ZIF-8 and IL@ZIF-8 samples, in agreement with the XRD data collected in this work. The intensities of the Bragg peaks are reduced for the $a_g$(IL@ZIF-8-LT) sample (see Supplementary Fig. 34) but it is clear that the sample has not been fully amorphized. As expected, the $S(Q)$s for the IL and $a_m$ZIF-8 samples show a noticeable absence of Bragg scattering, indicating their amorphous nature. The $S(Q)$ for the $a_g$(IL@ZIF-8-HT) sample is most similar to the $S(Q)$ of $a_m$ZIF-8, both of which have a broad first sharp diffraction peak (FSDP) often described as a manifestation of intermediate-range order in glasses[58,59]. In addition to this broad peak at ~1 Å$^{-1}$ in the $S(Q)$ of $a_g$(IL@ZIF-8-HT), there is a small sharper peak at 0.52 Å$^{-1}$. This matches the scattering vector observed for the (011) Bragg peak in the PXRD of ZIF-8.

From the background-corrected X-ray total scattering data, the real space PDFs, $G(r)$ could be extracted by Fourier transform and were subsequently converted to $D(r)$ in order to emphasize the peaks at high $r$[60–62]. These peaks correspond to atom–atom correlations in the sample, with the peak position determined by the interatomic distances between atom pairs and the intensity proportional to the product of the scattering factors from all of the atoms which correspond to a particular interatomic distance. As such, PDF analysis is a powerful tool for studying amorphous materials and glasses as the local short-range order can still be probed, but the presence of disorder and loss of structural coherence at longer length scales leads to the absence of peaks at this extended regime.

As the X-ray atomic form factor is proportional to the atomic number, $Z$, the relatively light elements such as those in the IL component do not contribute as intensely as correlations with the heavier zinc atoms in ZIF-8. Additionally, the IL only accounts for a theoretical 35 wt% of the composite sample, so the overall contribution of the IL towards the PDF pattern of the IL@ZIF-8 composite is expected to be small. Even so, a comparison of the PDFs of the IL@ZIF-8 composite, along with its constituent components in Fig. 8a, shows that for correlations in which there are peaks in both the IL and ZIF-8 patterns (e.g., the peaks at ~1.4 Å), the intensity of the corresponding peaks in the IL@ZIF-8 composite is greater than in the pure ZIF-8 pattern. For peaks which only correspond to correlations in ZIF-8 (e.g., the peaks at 2.01, 3.01, 4.17, and 6.02 Å), the intensity is smaller than in pure ZIF-8 due to the reduced ZIF-8 content in the composite.

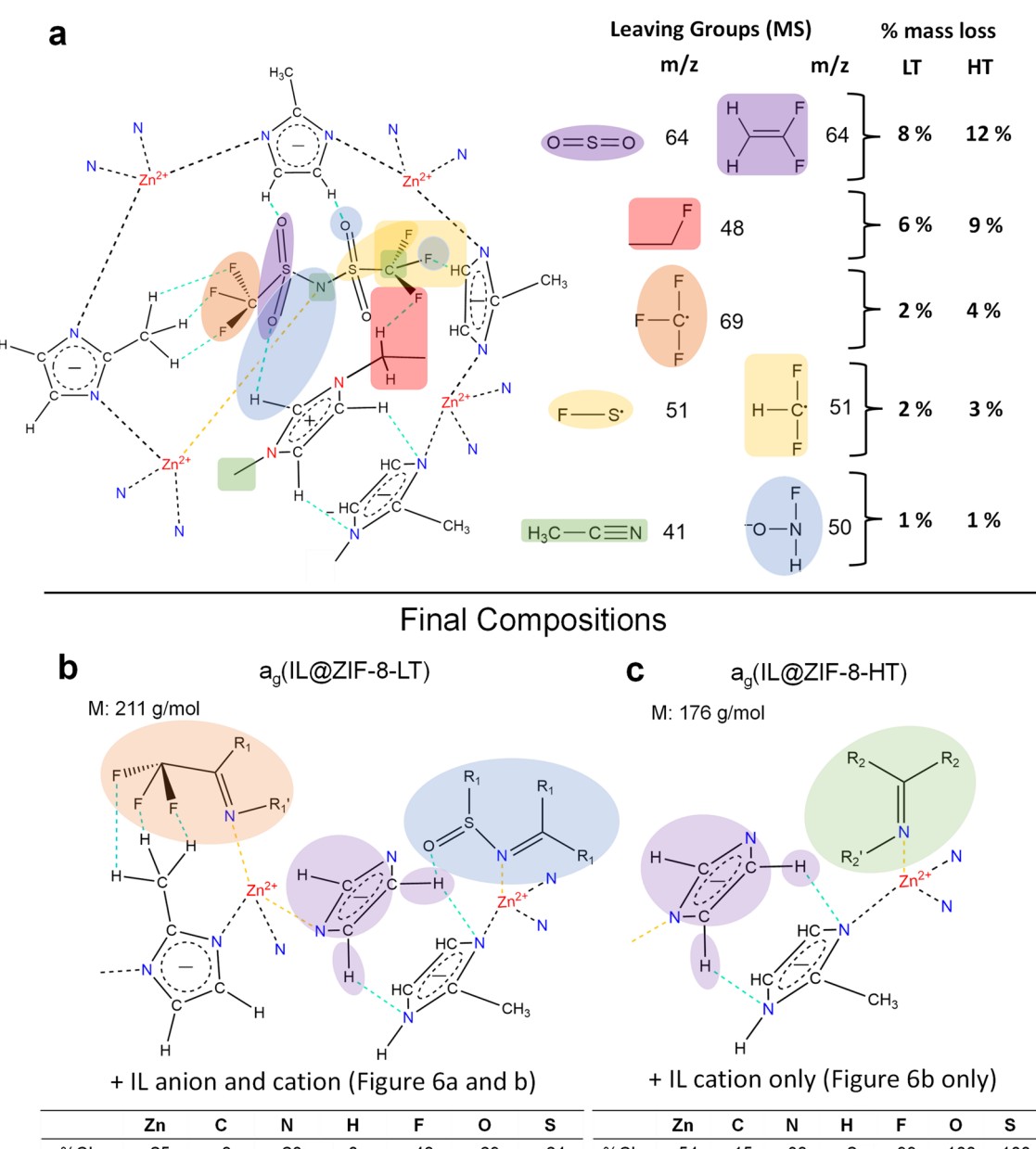

**Fig. 7 Route of decomposition and possible final compositions. a** Likely leaving groups and their percentages of the total mass loss observed from mass spectrometry. Detected masses <1 wt% loss are not included. Possible final compositions of **b** $a_g$(IL@ZIF-8-LT) and **c** $a_g$(IL@ZIF-8-HT) as determined from the peak area of the MS curves. Decomposition products are highlighted. $R_1$ is H/F or $CH_{3-x}F_x$ and $R_1'$ is only $CH_{3-x}F_x$ and, while $R_2$ is only an H-containing organic group, H or $CH_3$ and $R_2'$ is only $CH_3$. For a schematic view of the whole pore, see Supplementary Fig. 28.

Long-range order was evident in the ZIF-8 and IL@ZIF-8 samples, with peaks in the $D(r)$ extending out to 25 Å (see Supplementary Fig. 34b). However, the $D(r)$ for the $a_g$(IL@ZIF-8-HT) sample appears largely featureless at extended distances (>6.02 Å) which would be consistent with the vitrification of the ZIF-8 component. This loss of long-range order, alongside the retention of the local structure (short-range order), is consistent with glass formation, as has been observed for other glass-forming MOFs[37]. The retained short-range order is very similar to that exhibited by the pure ZIF-8 sample (see Fig. 8b), suggesting that the secondary building block units (i.e., the Zn(2-MeIm)$_4$ clusters) of ZIF-8 are still intact, though, their arrangement at extended length scales is disrupted. The presence of this

short-range order also suggests that the sample has not completely decomposed due to the heating procedure, or due to beam damage, consistent with the digestive [1]H NMR data. Finally, the PDF patterns of the $a_g$(IL@ZIF-8-HT) sample and the $a_m$ZIF-8, shown in Fig. 8b are remarkably similar which, together with the $S(Q)$ only showing broad scattering features and very weak remnant Bragg scattering (Supplementary Fig. 34a), provides further evidence for the almost complete amorphization of the ZIF-8 framework within the composite.

We conducted further investigations on the physical properties of the $a_g$(IL@ZIF-8-HT) glass by examining optical absorbance as well as mechanical properties. Absorbance spectra taken of an $a_g$(IL@ZIF-8-HT) glass film on a platinum substrate are presented

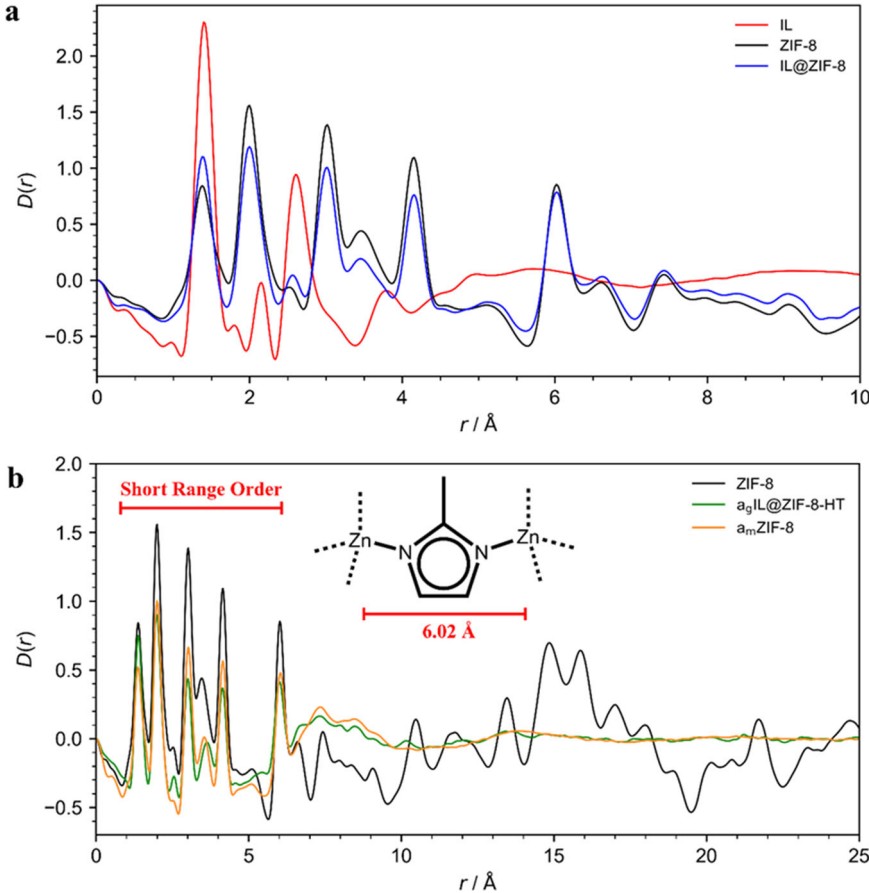

**Fig. 8 Ambient temperature X-ray pair distribution function data. a** Comparison of the X-ray pair distribution functions, D(r) of IL@ZIF-8 with its constituent components. **b** Comparison of the X-ray pair distribution functions, D(r) of $a_g$(IL@ZIF-8-HT), ZIF-8, and $a_m$ZIF-8. The limit of short-range order is shown along with the chemical connectivity of the ZIF-8 structure which accounts for the correlations within this limit.

in Supplementary Fig. 35 (experimental details are provided as Supplementary Information). The $a_g$(IL@ZIF-8-HT) glass is generally optically transparent but absorbing in the blue to the green part of the visible spectrum. This results in the brownish-red coloration observed in the microscope images presented in Fig. 3. Interestingly, the absorbance decreases towards the UV region, in the spectral range of ~500–360 nm (limit of the detector), which is a clear indicator for the absence of light scattering by crystal inclusions or other heterogeneities with a characteristic size corresponding to the wavelength of observation. Coloration aside, the obtained glass samples are highly transparent and possess no evidence of large-scale phase separation.

The mechanical properties of the $a_g$(IL@ZIF-8-HT) glass were studied through instrumented indentation testing using a nanoindenter (see Methods for details). The obtained depth profiles of hardness ($H$) and elastic modulus ($E$) are depicted in Supplementary Fig. 36. The indentation response displays a marked load dependence, which reflects in a continuous decrease of $H$ and $E$ with increasing displacement into the glass' surface ($h$). This effect is more pronounced at small indentation depths and tends to level off as the indenter penetration increases, with the values of $E$ and $H$ eventually approaching the values of crystalline ZIF-8 reported in the literature ($H = 0.501$ GPa and $E = 3.10$ GPa)[63]. Averaging the values of $E$ and $H$ between the upper 10% and lower 20% of each indentation depth profile resulted in mean values of $H = 0.730 \pm 0.136$ GPa and $E = 5.42 \pm 1.10$ GPa for $a_g$(IL@ZIF-8-HT). In a direct comparison with crystalline ZIF-8, we must emphasize that the mechanical

stability may strongly vary as a function of the employed synthesis conditions: values of $H$ ranging from 0.501 to 0.643 GPa have been reported for crystalline ZIF-8. Likewise, $E$ was reported with values of only 3.10 GPa up to about 7.33 GPa[63,64]. In this regard, the present $a_g$(IL@ZIF-8-HT) falls in the same category as other ZIF-type glasses, such as $a_g$ZIF-4 ($H = 0.676$ GPa and $E = 7.17$ GPa) or $a_g$ZIF-62 ($H = 0.656$ GPa and $E = 6.85$ GPa)[65].

## Discussion

In summary, the incorporation of an IL into the pores of ZIF-8 makes this otherwise unmeltable compound meltable. Based on structural investigations, we hypothesize that melting is achieved by reducing the melting temperature of ZIF-8 to below its thermal decomposition temperature, using electrostatic interactions of the IL (or partially decomposed IL fragments) at the ZIF-8 pores' interior surface in order to stabilize the rapidly dissociating ZIF-8 linkers upon heating. Although the methyl in mIm (ZIF-8 linker) likely changes the $Zn^{2+}$-linker bonding in ZIF-8 relative to meltable ZIF-4 (where only Im is present), the largest change is in topology: the sodalite ZIF-8 topology is over three times as porous as the cag ZIF-4 topology[18]. The IL anion and cation interact strongly with the $Zn^{2+}$ and linker, respectively, and the IL is less constrained than the ZIF-8 linker; upon infiltration, it decreases the internal surface area of ZIF-8 (see Supplementary Fig. 32 and Supplementary Table 9 for surface area and pore volume analysis). Thus, the high porosity of ZIF-8, or the low mobility and isolation of its linkers no longer present a barrier to melting: the IL ions are immediately available to exchange with the rapidly dissociating $Zn^{2+}$-linker bond and prevent decomposition at elevated

temperatures. We note that similar hydrophobicity and high thermal stability of the IL are important factors determining the meltability of IL-incorporated MOF composites. More generally, we conclude that a suitable, i.e., strongly bonding, IL stabilizes ZIF (and potentially other MOF) melts by reducing the lifetime of unstable configurations via $Zn^{2+}-N^-$ bonding and H bonding. The corresponding melt-quenched glass has a glass-forming ability which exceeds even those of previously reported super-strong glasses from conventionally meltable ZIFs.

This approach strongly broadens the variety of hybrid glass chemistries which may be derived from the MOF family. It offers exciting opportunities to melt other non-meltable crystalline MOFs, potentially enabling a broad range of hybrid glasses with a variety of physicochemical properties and corresponding applications, in particular, ones which are derived from MOF architectures with large pore sizes.

## Methods

**Preparation of IL@ZIF-8 composite**. ZIF-8 and the IL, 1-ethyl-3-methylimidazolium bis(trifluoromethylsulfonyl)imide, [EMIM][TFSI] (>99%), were purchased from ACSYNAM Inc and IoLiTec, respectively. To remove moisture and volatile impurities, ZIF-8 was evacuated at 105 °C under vacuum overnight. IL@ZIF-8 composite was prepared using wet impregnation at a stoichiometric IL loading of 35 wt%, according to previous reports. Briefly, 0.35 g of IL was dissolved in 20 mL acetone and stirred for 1 h at room temperature in a sealed container to hinder acetone evaporation. Afterward, activated ZIF-8 (0.65 g) was added to the solution and the mixture was stirred at 35 °C for about 7 h under an open atmosphere until the acetone was evaporated. The resultant powder sample was dried overnight at 105 °C to remove the remaining acetone.

**Glass samples**. Approximately 25 mg of powder IL@ZIF-8 composite was placed in a platinum crucible and pressed by hand to provide better contact with the crucible. To facilitate an even heat transfer in the sample, a smaller platinum crucible that could fit inside the sample crucible was placed on the sample. To obtain $a_g$(IL@ZIF-8-LT) and $a_g$(IL@ZIF-8-HT), the sample was heated to 120 °C with a ramp rate of 20 °C · min⁻¹ and kept for 45 min, followed by heating to 387 and 390 °C with a ramp rate of 10 °C · min⁻¹ and kept for 30 and 40 min, respectively. Afterward, the sample was cooled down to room temperature with a ramp rate of 50 °C · min⁻¹. All heating and cooling steps were performed under nitrogen flow (20 mL·min⁻¹).

**Washing experiment**. Approximately 40 mg of $a_g$(IL@ZIF-8-HT) and $a_g$(IL@ZIF-8-LT) were powdered gently and placed individually in 50 ml containers with 5 ml of acetone. Containers were covered with parafilm to hinder acetone evaporation during washing. Using a magnetic stirrer, washing was done at 50 °C for 4 h. After this step, washed $a_g$(IL@ZIF-8-HT), $a_g$(IL@ZIF-8-LT), and the filtrate were separated using filter paper. Collected washed $a_g$(IL@ZIF-8-HT) and $a_g$(IL@ZIF-8-LT) samples were dried in an oven at 80 °C for 4 h to remove residual acetone. To enhance the concentration of washed species in the filtrate, some of the filtrates were evaporated at 60 °C to a residue of 0.5 ml for further qualitative analysis.

**X-ray diffraction (XRD)**. A Rigaku MiniFlex diffractometer (Cu Kα X-ray source with a wavelength of 1.54059 Å) was used to collect diffractograms in the 2θ range of 5 to 40° with a step size of 0.02°. The voltage and current of the X-ray tube were set to 40 kV and 15 mA, respectively.

**Ambient temperature X-ray pair distribution function (XPDF)**. Synchrotron X-ray total scattering data were measured at the Diamond Light Source, UK (EE20038). Samples were hand-ground and loaded into borosilicate capillaries with a 1.17 mm inner diameter. The ZIF-8 and crystalline IL@ZIF-8 samples required the use of a beam filter due to detector saturation, giving a transmission factor of 0.519; all other samples were used without this beam filter. Data were collected for an empty capillary (used as a background) and for all samples to a $Q_{max}$ of 25.0 Å⁻¹ ($\lambda = 0.161669$ Å, 76.69 keV) with a collection time of 10 min per sample. Data normalization, background subtraction, and subsequent Fourier transform was performed using the GudrunX program to obtain the PDFs for each sample[60,61]. The atomic compositions used for this analysis were calculated from TG–MS data for the $a_g$(IL@ZIF-8-LT) and $a_g$(IL@ZIF-8-HT) samples.

**Amorphization of ZIF-8 via ball-milling**. About 50 mg of ZIF-8 was loaded into a 10 mL stainless steel jar with 2 mm × 7 mm stainless steel ball bearings. The jar was then placed into a Retsch MM400 grinder mill operating at 30 Hz for 30 min. The successful amorphization was confirmed by powder XRD.

**Fourier transform infrared (FTIR) spectroscopy**. FTIR spectra were collected using a Thermo Scientific Nicolet iS10 model FTIR spectrometer in attenuated total reflection mode. Sixty-four and 128 scans were measured for background and sample spectra with 2 cm⁻¹ resolution. Evaluation of the spectra was done using Fityk software[66].

**Thermogravimetric analysis coupled with differential scanning calorimetry (TGA-DSC)**. Thermogravimetric analysis (TGA) and DSC analysis were performed using a Netzsch STA 449 F1 instrument. Approximately 15 mg of each sample was placed in a platinum crucible and gently pressed by hand to ensure good contact between the crucible and the powder sample. All measurements were performed under 20 mL·min⁻¹ of nitrogen flow. First, the sample was heated to 120 °C with a ramp of 20 °C · min⁻¹ and equilibrated for 4 h to remove any volatiles. Subsequently, it was heated to 600 °C with a ramp rate of 5 °C · min⁻¹. To obtain the glass transition temperature ($T_g$), $a_g$(IL@ZIF-8-HT) in powder form was placed in a platinum crucible and heated to 400 °C with a 5 °C · min⁻¹ ramp rate. $T_m$, $T_d$, and $T_g$ are determined as the intersection of the starting baseline and the tangent to the DSC curve at the maximum gradient point.

**Cyclic $C_p$ scan**. A Netzsch STA 449 F1 instrument was used to perform cyclic $C_p$ scans. In these, the sample was heated up to 360 °C, then cooled to 200 °C, and heated again to 360 °C. All experiments used the identical heating rate of 20 °C·min⁻¹. The cooling rate between the first and the second upscan was varied from 10 °C min⁻¹ to 15 °C min⁻¹ and 20 °C min⁻¹. For each cycle, a baseline and a sapphire reference scan were recorded prior to the sample scan using the same temperature program. A platinum crucible was used for each cycle with a sample mass of around 15–20 mg. For each cycle, a new sample of IL@ZIF-8 was produced by placing the starting batch into the same Pt crucible as used for the subsequent DSC experiment and melting using the HT condition with a cooling rate of 5 °C·min⁻¹.

**Thermogravimetric analysis coupled with mass spectrometry (TG–MS)**. TG–MS analysis was performed using Netzsch STA 449 F1/QMS 403 instrument with multiple ion detection (MID) mode. Approximately 15 mg of each sample was placed in a platinum crucible and TG–MS analysis was performed for the samples at LT (387 °C for 30 min) and HT (390 °C for 40 min) conditions under 20 mL·min⁻¹ of nitrogen flow.

**Scanning electron microscopy (SEM)**. A JSM-7001 F microscope (Jeol Ltd, Japan) was used to analyze the morphology of ZIF-8 and IL@ZIF-8 samples. Approximately 10 mg of each sample was placed on a carbon tape and pasted on an aluminum cell. Prior to measurement, samples were coated with a thin layer of carbon. Voltage and working distance were set to 20 kV and 14 mm, respectively.

**Digital optical microscopy**. Imaging the glass samples was done using a Keyence VHX-6000 digital microscope with VHX-H2MK software and VHX-500 3D viewer 1.02. A VH-Z100UR differential interference contrast lens was used and the images were created by focal scanning along the z-axis and stacking images. Top lights with a side-lit lighting configuration was used to capture the photos with variable magnifications (200X, 250X, and 300X).

**Confocal laser scanning microscopy**. Melting of IL@ZIF-8 composite was recorded using a Carl Zeiss Axio imager-Z1m LSM700 confocal LSM. ZIF-8 and IL@ZIF-8 powders were placed on a quartz disc (5 mm diameter) and placed in a Linkam T95-HT stage. Samples were heated to 390 °C with a ramp rate of 10 °C · min⁻¹ under 18 mL·min⁻¹ of argon flow. Images were captured in 1 min intervals and ZEN-black software was used to create videos from captured images.

**Nuclear magnetic resonance spectroscopy (¹H NMR)**. A Bruker 300 MHz spectrometer was used to measure ¹H NMR spectra. Approximately 6 mg of each sample was digested in 0.7 mL of a stock solution of DCl (20%)/D₂O (0.889 mL) and DMSO-$d_6$ (3 mL). Data analysis was performed in TopSpin software. Predicted ¹H NMR spectra of decomposed IL structures was generated using www.nmrdb.org after drawing the corresponding chemical structure of decomposed EMIM cations[67,68].

**Solid-state nuclear magnetic resonance (SSNMR) spectroscopy**. Single-pulse ¹³C, as well as ¹H-¹³C and ¹H-¹⁵N CP SSNMR, was done using a Bruker Avance III 400 (9.4 T magnet, 400.17 MHz for ¹H, 100.62 MHz for ¹³C, and 40.55 MHz for ¹⁵N) equipped with a 4 mm MAS probehead. Carbon and nitrogen spectra were referenced to the external standard α-glycine (carbonyl peak at 176.5 ppm and ¹⁵N peak at 32.9 ppm, respectively). For single-pulse ¹³C experiments, relaxation times were varied from 10 − 150 s for the crystalline IL@ZIF-8 sample and the intensity for the longest relaxing carbon (C without any H) was found to be invariant for D1 > 100 s (t₁ ≈ 20 s); this relaxation time was used for all the single-pulse ¹³C experiments, assuming that glassy and IL carbons would relax faster. A 90° pulse of 10 µs (25 kHz) was used for single-pulse ¹³C, as were optimized on adamantane previously, spinning at 10 kHz, while the number of scans varied from 800 to 4104,

depending on the sample. Finally, for the collection of the IL spectrum, no spinning was used, a 90° pulse of 3.5 µs (71 kHz) was used with SPINAL-64 decoupling was also used at an rf field of 49 kHz.

For $^1$H-$^{13}$C CP, a 2.4 µs (104 kHz) $^1$H 90° pulse was used with a contact time of 2 ms with a recycle delay of 2.5 s at a $^{13}$C rf field of 71 kHz whilst the $^1$H rf field amplitude was ramped up to a maximum of 81 kHz. The number of scans varied from 2300 to 68,000 for ZIF-8 vs. a$_g$(IL@ZIF-8-HT). In the case of $^1$H-$^{15}$N CP, a 2.5 µs (100 kHz) $^1$H 90° pulse was used with a contact time of 3 ms with a recycle delay of 2.5 s at a $^{15}$N rf field of 56 kHz whilst the $^1$H rf field amplitude was ramped up to a maximum of 68 kHz. The number of scans varied from 24,000 to 168,000 for ZIF-8 vs. a$_g$(IL@ZIF-8-LT). Spinal-64 decoupling was applied with an rf field strength of 104 and 100 kHz for $^{13}$C and $^{15}$N, respectively. $^{13}$C was collected spinning at 10 kHz, while $^{15}$N was spun at 12 kHz.

**Gas sorption and Brunauer–Emmet–Teller (BET) analysis**. An Autosorb iQ instrument (Quantachrome) was used for BET surface area and pore volume analysis of ZIF-8, IL@ZIF-8, a$_g$(IL@ZIF-8-LT), and a$_g$(IL@ZIF-8-HT) samples. To quantify the BET surface area N$_2$ adsorption was monitored at 77 K. Around 50 mg of sample material were used for each measurement. Samples were outgassed for 20 h at $10^{-8}$ mbar, 125 °C prior to measurement. For the relatively low surface areas for a$_g$(IL@ZIF-8-LT) and a$_g$(IL@ZIF-8-HT) samples, N$_2$ and CO$_2$ adsorption experiments at 77 and 273 K, respectively, were repeated using adjusted parameters (equilibrium times of 100 s, pressure tolerance of 2% with respect to P$_0$, and degassing temperature of 150 °C).

**Nanoindentation tests**. Instrumented indentation experiments were carried at room temperature using a nanoindenter (G200, KLA Co.) equipped with a Berkovich diamond tip (Synton-MDP Inc.). Samples were mounted in a thermosetting epoxy resin (Araldite CY212, Agar Scientific Ltd.). The tip area function and the instrument's frame compliance were calibrated prior to the first experiments on a fused silica reference glass (Corning Code 7980, Corning Inc.), following the method of Oliver and Pharr[69]. Depth profiles of hardness $H$ and modulus $E$ were obtained from indentations conducted in the continuous stiffness mode[70]. In total, 25 indents with a depth limit of 2000 nm were performed per sample at a strain rate of 0.05 s$^{-1}$. Values of $H$ were calculated from the load divided by the projected contact area of the indenter tip and values of $E$ were derived from the reduced modulus[71].

## Data availability
The data that support the findings of this study are shown in the manuscript or supporting information, or available from the corresponding author on any request.

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

## Acknowledgements

This project has received funding from the European Research Council (ERC) under the European Union's Horizon 2020 research and innovation program (ERC grant no. 681652), the Carl Zeiss Foundation (Durchbrüche 2019), and the Deutsche Forschungsgemeinschaft (DFG, German Research Foundation) under Germany's Excellence Strategy—EXC 2051—Project-ID 390713860. T.D.B. acknowledges the Royal Society for a University Research Fellowship (UF150021), the Leverhulme Trust for a Philip Leverhulme Prize, and the University of Canterbury Te Whare Wānanga o Waitaha, New Zealand, for a University of Cambridge Visiting Canterbury Fellowship. J.M.T. acknowledges funding from the NanoDTC EPSRC Grant EP/L015978/1. We extend our gratitude to Diamond Light Source, Rutherford Appleton Laboratory, UK, for access to Beamline I15-1 (EE20038), Michael F. Thorne and Lauren N. McHugh at the University of Cambridge, for assistance in carrying out room temperature XPDF measurements, as well as Aaron Reupert, Rene Limbach, and Erik Troschke at the University of Jena, for technical assistance with laser scanning microscopy, indentation testing, and gas adsorption experiments, respectively.

## Author contributions

L.W., V.N., and T.D.B. conceived of this study. Sample synthesis, preparation, thermal analysis, and optical spectroscopy were done by V.N. C.C. conducted NMR studies. X-ray total scattering measurements and interpretations were carried out by J.M.T., with the assistance of D.A.K. and T.D.B. Microscopic investigations were performed by V.N., C.C., and L.W. V.N. and C.C. wrote the first draft of the manuscript with the help of L.W. All authors were involved in data interpretation, critical discussion, and manuscript revisions.

## Funding

## Competing interests

The authors declare no competing interests.

## Additional information

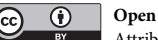

