## [Peer Review File · Nature Communications]

Ionic Liquid Facilitated Melting of the Metal-Organic Framework ZIF-8REVIEWER COMMENTS

Reviewer #1 (Remarks to the Author):

In this study, the ionic liquid [EMIM][TFSI] is incorporated into the pores of the MOF ZIF-8, $Zn(mIm)_2$, to create a composite that melts prior to decomposition. However, TGA and NMR evidence show that the IL molecules themselves degrade during the melting process, and the final chemical composition of the glassy material is not easy to determine. The lowering of the melting point of the composite compared to pristine ZIF-8 is hypothesized to be due to the ionic liquid stabilizing free linkers through supramolecular interactions after they dissociate from the zinc ions. Although the final material is indeed quite complex, the authors use several material characterization techniques to interrogate the structure and composition of the glass. I recommend this paper for publication after several important points are addressed.

1) The DSC in Figure 2C is only showing the heating experiment, for the IL@ZIF-8 and ag(IL@ZIF-8) and the T_m and T_g for both materials are small in nature. Running a heat/cool/heat experiment would show if the melting and glass transition are reversible and stable, not just noise in the traces. See Das, C.; Horike, S. Crystal melting vitrification behaviors of a three-dimensional nitrile-based metal organic framework. *Faraday Discuss.* 225, 403-413 (2021).

2) Throughout the article, terms need to be defined and discussed clearly. The authors use the term 'glass' to describe both their low temperature and high temperature materials; however, by PXRD the LT shows clear crystallinity so the definition of 'glass' is not clear. Additionally, they claim that the IL components interact strongly with the MOF components, effectively decreasing the melting temperature, but it seems more likely that decomposed (or decomposing) IL fragments are what stabilizes the amorphous MOF phase. Throughout the article the authors need to state whether they are referring to the pristine IL or decomposed IL fragments.

In addition to the above, we recommend some small changes to the text as follows:

The chemdraws to show supramolecular interactions need work. The angles and distances of many of the supramolecular interactions would not be feasible in reality. This reviewer suggests creating a figure with several panels to show portions of possible interactions rather than an entire ZIF-8 pore, in which the details become messy and unclear.

Many future applications of melted MOFs rely on accessible surface area. The melted MOF herein has extremely low surface area and this should be mentioned in the main text. The method of using ILs to stabilize the melting process is interesting and new, but the drawbacks of the current technique should be discussed as well.

In general, the authors place too much interpretation of their results in the Results section. Interpretation of the data and comparing the data to previous work should be in the Discussion section.

Reviewer #2 (Remarks to the Author):

This manuscript describes the effect of guest molecules as ionic liquids(IL) infiltrated into the pores of metal-organic framework, ZIF-8, on the melting behavior of IL@ZIF-8 composites. It is quite amazing to see that the IL incorporation decreased the melting temperature below the decomposition temperature. As the authors thoroughly investigated, the IL supports the dissociation of Zn-N coordination bond and stabilized the zinc ions and 2-methyl imidazole after the dissociation. On one hand, I totally understand the importance of this manuscript and the authors' logic as "the incorporated IL decreased the melting point of ZIF-8" as even highlighted in the title. On the other hand, I felt a bit uncomfortable with this statement. This

is most likely because I would recognize this as a new composite of "IL@ZIF-8" rather than the ZIF-8 itself. In this case, we cannot say that the IL decreased the melting point of ZIF-8 but rather say that the IL@ZIF-8 showed the lower melting point than the pristine ZIF-8. In particular, the corresponding glasses, which were formed by the thermal treatment at 387 C (LT) or 390 C (HT) followed by quick-quenching, have the decomposed IL molecules therein and thus, the different chemical compositions. If these decomposed IL molecules were removed from the pores and produced a new phase of pristine ZIF-8 glass, I would definitely agree with this argument as the IL supports the melting of ZIF-8. In my opinion, the reversibility of guest incorporation/removal is necessary to justify the authors' claim. I would potentially suggest the acceptance of this manuscript, but this argument and the following scientific concerns should be addressed before the official acceptance.

1. How did the authors decide the temperatures for the thermal treatment at 387 C and 390 C for the LT and HT samples, respectively? Why does the LT sample maintain a partial crystallinity even though the sample was treated at the higher temperature than the melting point (381 C)? Did the authors first try the variable temperature XRD to find the appropriate temperatures?

2. The LT sample looks interesting because it still maintains the crystallinity. However, the thermal analysis was not reported in Figure 2c and d. Please provide these data and discuss how it is different from the HT sample, which is totally amorphous.

3. The authors choose this specific IL, [EMIM][TFSI], because it is hydrophobic. However, after the careful analysis using the solid-state NMR, this IL molecules effectively interact with the ZIF-8 framework via hydrogen bonds. Did the authors try another IL that is hydrophobic but not strongly interact with the framework? In this manuscript, only one IL was used so that it is more difficult to really understand the effect of IL on the melting behavior.

4. In the Figure 5b, the data of the HT sample is missing. Was this crystalline peak disappeared in the HT sample?

Reviewer #3 (Remarks to the Author):

In this manuscript, Nozari, et al, incorporated organic ionic liquid into the pores of ZIF-8 which is a non-meltable MOF, expecting the formation of ZIF-8 liquid prior decomposition at elevated temperature and hence the formation of glass after quenching the molten. Attempts are then made to understand the melting and the glass formation processes of ZIF-8 loaded with ionic liquid using DSC, FTIR, NMR, PDF, and XRD measurements. In addition, they examined the porosity of the melt-quenched glass using N₂ gas adsorption measurement.

The idea of this manuscript may be good which will help to convert the non-meltable MOF crystals into glasses. But, I have several significant issues with this manuscript. My concerns are as follows:

1- The glass transition (@ 322oC, Fig. 2C) of the quenched glass ag(IL@ZIF-8-HT) is not clear! Cp trace is necessary to determine the accurate Tg and whether the observed endothermic peak is correlated with the amorphous MOF structure or it is due to the experimental error. Otherwise, the manuscript will lose its purpose and the melting of IL@ZIF-8 will be useless. In addition, the XRD and Cp measurements are needed for the heated ionic liquid at 390 oC to ensure the heated ionic liquid doesn't have amorphous or glassy nature.

2- Why did you measure the TGA trace of the ag(IL@ZIF-8-HT) glass only up to 400 0C (Fig. 2D)? consistency of curves is important.

3- The most significant concern is that—throughout the discussion/interpretation—I find consistently that the authors deeply focused on the melting mechanism of the IL@ZIF-8-HT, however, the investigation on the glassy nature of the composite sample is lacked. So, I recommend the authors to examine the influence of ionic liquid concentration on the glass formation process, transparency, and porosity (i.e. ZIF-

8 loaded by 5, 10, 15, 20, 25, and 30 wt%).

4- "It offers exciting opportunities to melt other non-meltable crystalline MOFs, enabling a broad range of new hybrid glasses" They didn't clearly confirm whether the melt-quenched sample possesses glassy nature.

5- What is the potential application of the ZIF-8 glass loaded with ionic liquid? What are the unique physicochemical properties of this MOF glass? The glass is almost opaque with very small pores volume ($0.001 \text{ cm}^3\text{g}^{-1}$) and it can't be prepared in large scale.

5- The authors state they obtained the MOFs glass. The glass samples they showed are grey or dark due to the decomposition. At least, glass samples should be transparent and exhibit clearly glass transition phenomenon.

6- We do not see any promising applications of such glasses.

Consequently, I do not recommend the acceptance of the paper for publication in the Nature Communications.

Reviewer Comments

Reviewer #1 (Remarks to the Author):

In this study, the ionic liquid [EMIM][TFSI] is incorporated into the pores of the MOF ZIF-8, Zn(mIm)₂, to create a composite that melts prior to decomposition. However, TGA and NMR evidence show that the IL molecules themselves degrade during the melting process, and the final chemical composition of the glassy material is not easy to determine. The lowering of the melting point of the composite compared to pristine ZIF-8 is hypothesized to be due to the ionic liquid stabilizing free linkers through supramolecular interactions after they dissociate from the zinc ions. Although the final material is indeed quite complex, the authors use several material characterization techniques to interrogate the structure and composition of the glass. I recommend this paper for publication after several important points are addressed.

Comment 1) *The DSC in Figure 2C is only showing the heating experiment, for the IL@ZIF-8 and a_g(IL@ZIF-8) and the T_m and T_g for both materials are small in nature. Running a heat/cool/heat experiment would show if the melting and glass transition are reversible and stable, not just noise in the traces. See Das, C.; Horike, S. Crystal melting vitrification behaviors of a three-dimensional nitrile-based metal organic framework. *Faraday Discuss.* 225, 403-413 (2021).*

Response. We appreciate the reviewer's comment. Accurate assignment of the glass transition is a challenge in these hybrid materials and the reviewer is absolutely right in that careful C_p scanning should be performed to avoid misinterpretations. As suggested by the reviewer, we now conducted a range of cycle C_p scans for a_g(IL@ZIF-8-HT) samples with altered temperature programs (up-/down-scan rates). Accordingly, we updated Figure 2, adding the cycle C_p scan using 20 °C·min⁻¹ as heating and cooling rate as Figure 2d, as shown below. Details of the C_p measurements are summarized in Methods section and highlighted in yellow (in short, the C_p scan is a referenced DSC scan).

Figure 2. Structural characterization, enthalpic responses, and cyclic C_p scan. (a) XRD patterns of ZIF-8, IL@ZIF-8 crystalline composite, $a_g(\text{IL@ZIF-8-LT})$ and $a_g(\text{IL@ZIF-8-HT})$ samples. Crystallographic data is taken from literature.²⁵ (b) FTIR spectra obtained for ZIF-8, IL, crystalline IL@ZIF-8 composite, $a_g(\text{IL@ZIF-8-LT})$, and $a_g(\text{IL@ZIF-8-HT})$. (c) DSC scans of ZIF-8, IL, IL@ZIF-8, and $a_g(\text{IL@ZIF-8-HT})$ samples with heating rate of $5\text{ }^\circ\text{C}\cdot\text{min}^{-1}$. T_m and T_d are indicated as offset temperature of melting peak and onset temperature of decomposition of IL@ZIF-8, respectively. T_g is defined as the onset temperature of glass transition peak of $a_g(\text{IL@ZIF-8-HT})$. (d) Cyclic C_p scan of $a_g(\text{IL@ZIF-8-HT})$ with heating and cooling rate of $20\text{ }^\circ\text{C}\cdot\text{min}^{-1}$.

Cycle C_p scans with different cooling rates are included in Supplementary information file as Supplementary Figure S4, as shown below:

Supplementary Figure S4. Cyclic DSC (C_p) scan of $a_g(\text{IL@ZIF-8-HT})$ using different cooling rates between two consecutive heating scans. **(a)** Cooling rate of $15^{\circ}\text{C}\cdot\text{min}^{-1}$. **(b)** Cooling rate of $10^{\circ}\text{C}\cdot\text{min}^{-1}$. Insets show the respective heating and cooling rates.

We added the following text on page 7 and highlighted in yellow:

“To more clearly observe the glass transition, referenced DSC runs were conducted in order to extract the isobaric heat capacity C_p (denoted C_p scans). Cyclic C_p scans were performed on $a_g(\text{IL@ZIF-8-HT})$ samples at various heating and cooling rates (*see* methods section for further details). Figure 2d shows a cyclic C_p scan for $a_g(\text{IL@ZIF-8-HT})$ using $20^{\circ}\text{C}\cdot\text{min}^{-1}$ as heating and

cooling rate. A pronounced glass – liquid transition is detected at ~ 328 °C, with a configurational heat capacity ΔC_p of ~ 0.11 J·g⁻¹·K⁻¹ (**Figure 2d**). The magnitude of the jump in C_p is comparable to the values observed for other ZIF glasses such as ZIF-4, 0.11 and 0.16 J·g⁻¹·K⁻¹ for LDA and HDA phases, and 0.19 J·g⁻¹·K⁻¹ for ZIF-62.^{1,2} The minor up-shift in the glass transition as compared to the DSC scan (**Figure 2c**) is attributed to the higher scanning rate (20 °C·min⁻¹ versus 5 °C·min⁻¹). Although the second upscan exhibits a similar ΔC_p to the first, a slight delay is found in the glass transition. We attribute this observation to the continuing interaction between the glass phase and residual IL, simultaneously overlapping with ongoing IL decomposition during extended exposure of the glass to high temperatures in the first upscan and subsequent down scan (which is why we limited the scanning range to the upper limit of 360 °C). Additional cycle C_p scans with different cooling/heating rates are provided in Supplementary Figure S4. For a_g(IL@ZIF-8-LT), shown in Supplementary Figure S5, the glass–liquid transition is not as clearly visible as in the HT-sample, which is because crystalline ZIF-8 (together with residual IL) remains the primary phase in this case, and neither ZIF-8 nor the IL exhibit a DSC feature in this temperature range (**Figure 2c**). However, in the second upscan, a weak glass transition is detected also for the LT sample, what is *in line* with our interpretation of progressive reactions during DSC scanning.’’

The methods section on page 26 is updated as well, and details of C_p measurements are provided as follows:

‘**Cyclic C_p scan.** A Netzsch STA 449 F1 instrument was used to perform cyclic C_p scans. In these, the sample was heated up to 360 °C, then cooled to 200 °C, and heated again to 360 °C. All experiments used the identical heating rate of 20 °C·min⁻¹. The cooling rate between the first and the second upscan was varied from 10 °C·min⁻¹ to 15 °C·min⁻¹ and 20 °C·min⁻¹. For each cycle, a baseline and a sapphire reference scan were recorded prior to the sample scan using the same temperature program. A platinum crucible was used for each cycle with a sample mass of around 15-20 mg. For each cycle, a new sample of IL@ZIF-8 was produced by placing the starting batch into the same Pt crucible as used for the subsequent DSC experiment, and melting using the HT condition with a cooling rate of 5 °C·min⁻¹.’’

We note that the TGA data in previous Figure 2d is moved to Supplementary Information as Supplementary Figure S2 and highlighted in yellow as shown below:

Supplementary Figure S2. Thermogravimetric analysis. (a) ZIF-8, IL, and IL@ZIF-8 composite. (b) $a_g(\text{IL@ZIF-8-HT})$. A heating rate of $5\text{ }^\circ\text{C}\cdot\text{min}^{-1}$ was used to obtain the TG-DSC scans. Corresponding DSC scans are shown in Figure 2c.

Regarding the T_m , as it is mentioned in the paper, the onset of melting of IL@ZIF-8 was detected at $381\text{ }^\circ\text{C}$, however, complete melting occurs once the material is heated for 40 minutes isothermally at $390\text{ }^\circ\text{C}$.

Comment 2) Throughout the article, terms need to be defined and discussed clearly. The authors use the term ‘glass’ to describe both their low temperature and high temperature materials; however, by PXRD the LT shows clear crystallinity so the definition of ‘glass’ is not clear. Additionally, they claim that the IL components interact strongly with the MOF components, effectively decreasing the melting temperature, but it seems more likely that decomposed (or decomposing) IL fragments are what stabilizes the amorphous MOF phase. Throughout the article the authors need to state whether they are referring to the pristine IL or decomposed IL fragments.

Response. We thank the reviewer for this excellent comment. On one occasion in the text where we referred to the LT sample as the melt-quenched glass, and we corrected from “and in the melt-quenched glasses, $a_g(\text{IL@ZIF-8-LT})$ and $a_g(\text{IL@ZIF-8-HT})$ ” to ‘the melt-quenched glass, $a_g(\text{IL@ZIF-8-HT})$, and the crystal-glass composite $a_g(\text{IL@ZIF-8-LT})$ ’ and highlighted in yellow on page 9.

According to our DSC and TGA data presented in Figure 2c and Supplementary Figure S2, melting of the IL@ZIF-8 occurs before reaching the decomposition temperature. This means that strong interaction between the IL and ZIF-8 is the major parameter leading to the melting. However, the reviewer is absolutely right in that complete melting and formation of the glass requires an isothermal step where some of the IL molecules partially decomposes. Based on this, we added “(or partially decomposed IL molecules)” and “(or partially decomposed IL fragments)” where we mention that the IL stabilizes the interactions on pages 16 and 23, respectively.

Comment: *In addition to the above, we recommend some small changes to the text as follows: The chemdraws to show supramolecular interactions need work. The angles and distances of many of the supramolecular interactions would not be feasible in reality. This reviewer suggests creating a figure with several panels to show portions of possible interactions rather than an entire ZIF-8 pore, in which the details become messy and unclear.*

Response. We again thank the reviewer for this comment. It is challenging to represent a complicated system with only chemdraws. We have done as the reviewer suggested – created cut-aways highlighting the important and unique interactions in Figure 6 and Figure 7. Additionally, we moved the whole pore diagram to the Supplementary Information. Figure 7a is an exception, where we felt that the whole pore and IL interaction needed to be represented to explain all the different leaving groups properly. Revised Figures 6 and 7 shown below are replaced with previous Figures.

Figure 6. 2D schematic of possible interactions between IL and ZIF-8. (a) Interactions with IL anion (b) Interactions with IL cation. (c) Melting/amorphization of IL@ZIF-8 at 381 °C. Orange and green bonds represent $Zn^{2+}-N^{-}$ bonding and H-bonding between IL and ZIF-8, respectively. For a diagram of the whole pore, *see* Supplementary Figure S27.

Figure 7. Route of decomposition and possible final compositions. (a) Likely leaving groups and their percentages of the total mass loss observed from mass spectrometry. Detected masses < 1 wt% loss are not included. Possible final compositions of (b) $a_9(\text{IL@ZIF-8-LT})$ and (c) $a_9(\text{IL@ZIF-8-HT})$ as determined from the peak area of the MS curves. **Decomposition products are highlighted.** R_1 is H/F or $\text{CH}_{3-x}\text{F}_x$ and R_1' is only $\text{CH}_{3-x}\text{F}_x$ and, while R_2 is only an H-containing organic group, H or CH_3 and R_2' is only CH_3 . **For a diagram of the whole pore, see Supplementary Figure S28.**

Comment: *Many future applications of melted MOFs rely on accessible surface area. The melted MOF herein has extremely low surface area and this should be mentioned in the main text. The method of using ILs to stabilize the melting process is interesting and new, but the drawbacks of the current technique should be discussed as well.*

Response. We thank reviewer for this very valuable comment. To investigate the porosity of the $a_g(\text{IL@ZIF-8-LT})$ and $a_g(\text{IL@ZIF-8-HT})$ samples, we performed additional gas adsorption measurements using N_2 at 77 K and CO_2 at 273 K, as well as washing experiments by which residual IL/IL fragments were partially removed from the IL@ZIF-8 glass. This comment is essentially equivalent to a comment of reviewer 2. Please refer further to our response to reviewer 2 and reviewer 3.

Comment: *In general, the authors place too much interpretation of their results in the Results section. Interpretation of the data and comparing the data to previous work should be in the Discussion section.*

Response. We appreciate reviewer's comment. However, we are unsure about the article guidelines of Nat. Commun., where the "Discussion" section is – in our understanding – "only" a short summary/conclusion and outlook paragraph, whereas the major part of the scientific interpretation and data evaluation is indeed placed in the "Results" section, and sub-sections are not commonly used in the journal. We would gladly adapt these sections, but some advice from the editor would be very helpful.

Reviewer #2 (Remarks to the Author):

This manuscript describes the effect of guest molecules as ionic liquids (IL) infiltrated into the pores of metal-organic framework, ZIF-8, on the melting behavior of IL@ZIF-8 composites. It is quite amazing to see that the IL incorporation decreased the melting temperature below the decomposition temperature. As the authors thoroughly investigated, the IL supports the dissociation of Zn-N coordination bond and stabilized the zinc ions and 2-methyl imidazole after the dissociation. On one hand, I totally understand the importance of this manuscript and the

authors' logic as "the incorporated IL decreased the melting point of ZIF-8" as even highlighted in the title. On the other hand, I felt a bit uncomfortable with this statement. This is most likely because I would recognize this as a new composite of "IL@ZIF-8" rather than the ZIF-8 itself. In this case, we cannot say that the IL decreased the melting point of ZIF-8 but rather say that the IL@ZIF-8 showed the lower melting point than the pristine ZIF-8. In particular, the corresponding glasses, which were formed by the thermal treatment at 387 C (LT) or 390 C (HT) followed by quick-quenching, have the decomposed IL molecules therein and thus, the different chemical compositions. If these decomposed IL molecules were removed from the pores and produced a new phase of pristine ZIF-8 glass, I would definitely agree with this argument as the IL supports the melting of ZIF-8. In my opinion, the reversibility of guest incorporation/removal is necessary to justify the authors' claim. I would potentially suggest the acceptance of this manuscript, but this argument and the following scientific concerns should be addressed before the official acceptance.

Response. We are absolutely with the reviewer; this is an important comment. As suggested by the reviewer we adapted the manuscript in two regards: when referring to the $a_g(\text{IL@ZIF-8-HT})$ glass, we point-out that this is a composite glass, not simply a "ZIF-8 glass" (the LT material is a crystal-glass composite in itself). More importantly, we investigated the removal of unreacted/decomposed IL from the $a_g(\text{IL@ZIF-8-HT})$ and $a_g(\text{IL@ZIF-8-LT})$ samples by washing the samples using acetone or DMSO as solvents (whereby the use of acetone turned out to be preferential in terms of observing the results of washing before/after using IR-ATR spectroscopy on the washed sample and on the washing solutions, respectively). After washing, we first investigated the qualitative optical appearance of the samples using a digital microscope as mentioned in the methods section (whereby washed samples became significantly brighter). Then, N_2 and CO_2 gas adsorption measurements were conducted on samples before and after washing (whereby a significant, *i.e.*, fourfold enhancement of total porosity was detected, taken as evidence for the removal of IL residues). Moreover, a DSC-TG experiment was performed on post-washed $a_g(\text{IL@ZIF-8-HT})$ to demonstrate glassy nature of the washed sample and the persistent occurrence of a glass transition. Details of washing experiment are included in the Methods section as follows:

Washing experiment. Approximately 40 mg of $a_g(\text{IL@ZIF-8-HT})$ and $a_g(\text{IL@ZIF-8-LT})$ were powdered gently and placed individually in 50 ml containers with 5 ml of acetone. Containers were covered with parafilm to hinder acetone evaporation during washing. Using a magnetic stirrer, washing was done at 50 °C for 4 hours. After this step, washed $a_g(\text{IL@ZIF-8-HT})$, $a_g(\text{IL@ZIF-8-}$

LT) and the filtrate were separated using filter paper. Collected washed $a_g(\text{IL@ZIF-8-HT})$ and $a_g(\text{IL@ZIF-8-LT})$ samples were dried in an oven at 80 °C for 4 hours to remove residual acetone. To enhance the concentration of washed species in the filtrate, some of filtrates were evaporated at 60 °C to a residue of 0.5 ml for further qualitative analysis.”

FTIR spectra were collected for $a_g(\text{IL@ZIF-8-HT})$ and $a_g(\text{IL@ZIF-8-LT})$ prior to washing, on washed $a_g(\text{IL@ZIF-8-HT})$ and $a_g(\text{IL@ZIF-8-LT})$, and on the filtrate obtained after washing. Results are shown and discussed in the Supplementary Information as Supplementary Figure S29–S32:

“We investigated the removal of unreacted/decomposed IL from the $a_g(\text{IL@ZIF-8-HT})$ and $a_g(\text{IL@ZIF-8-LT})$ samples by washing the samples using acetone as the solvent. Details on the washing procedure are provided in the Methods section. In the IR spectra of the filtrates obtained from washing $a_g(\text{IL@ZIF-8-LT})$ and $a_g(\text{IL@ZIF-8-HT})$ samples, *see* Supplementary Figure S29, new bands are detected in the highlighted regions of the filtrates as compared to clean acetone, indicating partial uptake of soluble IL-related compounds from the $a_g(\text{IL@ZIF-8-LT})$ and $a_g(\text{IL@ZIF-8-HT})$ samples. Band positions of the newly emerged peaks in the filtrate are in the same regions where peak intensities/positions are different in the washed spectra. Microscope images of the washed $a_g(\text{IL@ZIF-8-LT})$ and $a_g(\text{IL@ZIF-8-HT})$ are presented in Supplementary Figure S30. They reveal smooth glassy surfaces with sharp edges, indicating that the samples remained stable during the washing experiment. Removal of some part of the decomposed IL is also evident from the color of the washed samples, which become notably clearer as compared to their appearance before washing (Figure 3 of the main text). Correspondingly, the filtrates (Supplementary Figure S30c) exhibit a yellow-brownish tint, which we take as further evidence for the washing-off of at least a part of the decomposed species from $a_g(\text{IL@ZIF-8-LT})$ and $a_g(\text{IL@ZIF-8-HT})$ samples. The glassy nature of washed $a_g(\text{IL@ZIF-8-HT})$ was examined by performing DSC-TGA cyclic experiments as shown in Supplementary Figure S31. A clear glass transition was observed on both upscans, however, due to extended exposure of the glass to high temperature at the first upscan, a shift of T_g to higher temperature ($\sim +5$ °C) occurred on the second upscan.

Furthermore, we conducted N_2 and CO_2 adsorption experiments on unwashed and washed $a_g(\text{IL@ZIF-8-HT})$ and $a_g(\text{IL@ZIF-8-LT})$ samples (washed samples are denoted as post-washing

“PW”). The N₂ and CO₂ adsorption isotherms presented in Supplementary Figure S32 show a notably enhanced gas uptake after the washing. The increase in total pore volume is about fourfold in both the N₂ and the CO₂ experiment as compared to unwashed a_g(IL@ZIF-8-HT) and a_g(IL@ZIF-8-LT). In terms of N₂ uptake at 77 K, the washed samples clearly outperform other MOF glasses such as a_gZIF-62 and a_gZIF-76-mbIm, where no uptake was observed at 77 K in those glasses.^{3,4} For CO₂ uptake at 273 K and 1 bar, a_g(IL@ZIF-8-HT) and a_g(IL@ZIF-8-LT) show 10 and 12 cc (STP) g⁻¹ which is lower than 18 cc (STP) g⁻¹ reported for a_gZIF-62.⁵ However, the washed a_g(IL@ZIF-8-HT)-PW and a_g(IL@ZIF-8-LT)-PW samples (with 24 and 29 cc (STP) g⁻¹, respectively) again outperform data reported for a_gZIF-62 and a_g[(ZIF-8)_{0.2}(ZIF-62)_{0.8}] (18.7 cc (STP) g⁻¹) at similar temperature and pressure.³”

Supplementary Figure S29. FTIR spectra of a_g(IL@ZIF-8-LT) and a_g(IL@ZIF-8-HT) samples prior and after washing with acetone at 50 °C. **(a)** a_g(IL@ZIF-8-LT) sample; **(b)** a_g(IL@ZIF-8-HT) sample.

Supplementary Figure S30. Microscope images of the post-washing $a_g(\text{IL@ZIF-8-LT})$ and $a_g(\text{IL@ZIF-8-HT})$ samples. (a) post-washing $a_g(\text{IL@ZIF-8-HT})$ sample; (b) post-washing $a_g(\text{IL@ZIF-8-LT})$ sample; (c) washing filtrates. Scale bars are 100 μm .

Supplementary Figure S31. DSC-TGA scan of washed $a_g(\text{IL@ZIF-8-HT})$ glass taken at $20\text{ }^\circ\text{C}\cdot\text{min}^{-1}$.

Supplementary Figure S32. Adsorption isotherms obtained for $a_g(\text{IL@ZIF-8-LT})$ and $a_g(\text{IL@ZIF-8-HT})$ samples and post-washing $a_g(\text{IL@ZIF-8-LT})\text{-PW}$ and $a_g(\text{IL@ZIF-8-HT})\text{-PW}$ samples. (a) N_2 isotherms at 77 K. (b) CO_2 isotherms at 273 K.

Supplementary Table S9. BET Surface area and pore volume results.

sample	Surface area (m ² g ⁻¹)	Pore volume (cm ³ g ⁻¹)
ZIF-8	1752	0.634
IL@ZIF-8	11	0.005
a_g(IL@ZIF-8-LT)	17	0.003
a_g(IL@ZIF-8-HT)	16	0.001
a_g(IL@ZIF-8-LT)*	17	0.021
a_g(IL@ZIF-8-HT)*	16	0.011
a_g(IL@ZIF-8-LT)-PW*	170	0.096
a_g(IL@ZIF-8-HT)-PW*	28	0.043
*Measured using optimized parameters. See Methods section for details.		

Based on these additional experiments we added the following paragraph in the main text on page 19:

“We investigated the removal of IL residues from the a_g(IL@ZIF-8-HT) and a_g(IL@ZIF-8-LT) samples by conducting post-process washing experiments in acetone. Details are provided in the Supplementary Information, Supplementary Figures S29–S32, and Supplementary Table S9 for the obtained results. In short, washing results in a significant reduction of the brownish tint (which we have assigned to the decomposition products) and notably enhanced gas adsorption performance by partially removing the unreacted or decomposed compounds from glass matrix. DSC-TGA scanning of post-washed a_g(IL@ZIF-8-HT), Supplementary Figure S31, confirmed that the glassy nature of the sample is preserved upon washing. Gas adsorption experiments using N₂ or CO₂ (Supplementary Figure S32) reveal a 4-fold increase of the glass’ total porosity after washing; in particular, the washed a_g(IL@ZIF-8-LT) sample which contains a major fraction of crystalline ZIF-

8 (being only partially amorphized) exhibited improved CO₂ uptake compared to the adsorption capacity observed for other MOF glasses such as a_gZIF-62 and a_g[(ZIF-8)_{0.2}(ZIF-62)_{0.8}].^{3,5} While outside of the scope of the present study, future exploration of LT-type composite materials containing crystalline ZIF-8 in a partially melted IL@ZIF-8 matrix phase may be beneficial towards application of these materials, for example, in gas adsorption and separation.”

BET analysis in the Methods section is also updated and highlighted in yellow.

“**Gas sorption and Brunauer-Emmet-Teller (BET) analysis.** An Autosorb iQ instrument (Quantachrome) was used for BET surface area and pore volume analysis of ZIF-8, IL@ZIF-8, a_g(IL@ZIF-8-LT), and a_g(IL@ZIF-8-HT) samples. To quantify the BET surface area N₂ adsorption was monitored at 77 K. Around 50 mg of sample material were used for each measurement. Samples were outgassed for 20 h at 10⁻⁸ mbar, 125 °C prior to measurement. For the relatively low surface areas for a_g(IL@ZIF-8-LT) and a_g(IL@ZIF-8-HT) samples, N₂ and CO₂ adsorption experiments at 77 K and 273 K, respectively, were repeated using adjusted parameters (equilibrium times of 100 s, pressure tolerance of 2% with respect to P₀ and degassing temperature of 150 °C).”

In summary, these new data clearly show that the residual IL can at least partially be removed from the quenched IL@ZIF-8 glass by washing in acetone. This washing process does not alter the glassy nature of the material, or dissolve the glass. It leads to very significantly enhanced material porosity, which is perfect starting point for future research, *e.g.*, regarding potential applications in gas separation or adsorption. At the present stage, we believe that it provides strong confirmation for the broad interest in using IL-assisted melting of MOFs to enhance the variety of MOF glasses.

Comment 1: *How did the authors decide the temperatures for the thermal treatment at 387 C and 390 C for the LT and HT samples, respectively? Why does the LT sample maintain a partial crystallinity even though the sample was treated at the higher temperature than the melting point (381 C)? Did the authors first try the variable temperature XRD to find the appropriate temperatures?*

Response. We thank the reviewer for this comment. IL@ZIF-8 composite melting starts at 381 °C, as can be seen from the melting peak in DSC, and this was also confirmed by *in-situ* microscopic observation, shown in the supplementary video 2. However, complete melting of the composite

occurs only after heating the sample isothermally for a given time; we observe that the melting reaction is slow (which is common in the melting of MOFs and inorganic zeolites, and is related to the viscosity of the obtained liquid phase, which is – being close to T_g – very high, and, eventually, to the kinetics of IL decomposition and IL-ZIF-8 interaction). Based on different heat-treatments which we conducted on IL@ZIF-8 and subsequent analysis using XRD presented in the Figure below, LT and HT conditions were selected to obtain partially-amorphized and fully-amorphized samples, respectively. We note here that the partially amorphized material might be particularly interesting for potential applications such as gas separation; however, a detailed analysis of the relation between process parameters and stabilization of a given fraction of crystalline ZIF-8 (by only partially melting) is not in the scope of this present study.

We included a Figure showing different heat-treatments in Supplementary Information as Supplementary Figure S3 and added the following text in the manuscript on page 7. ‘’This suggests that treatment temperature and time are important parameters in the formation of IL@ZIF-8 glasses and crystal-glass composite samples, as indicated in Supplementary Figure S3 for a range of tested synthesis conditions.’’

Supplementary Figure S3. X-ray diffraction patterns on IL@ZIF-8 composites following various heat-treatment conditions. Conditions for the LT and HT samples were selected from this initial screening procedure (as labelled).

Comment 2. *The LT sample looks interesting because it still maintains the crystallinity. However, the thermal analysis was not reported in Figure 2c and d. Please provide these data and discuss how it is different from the HT sample, which is totally amorphous.*

Response. The reviewer is absolutely right. Application-wise, the LT-route certainly deserves future attention because it enables fabrication of crystal-glass composites, potentially with a tailored crystalline fraction. However, it is clear that an in-depth study into this direction is outside of the scope of the present manuscript. For the moment, as suggested by the reviewer, we performed a cyclic C_p scan for the $a_g(\text{IL@ZIF-8-LT})$ sample and included in the Supplementary Information file as Supplementary Figure S5. The following text was added to the manuscript on page 8:

Supplementary Figure S5. Cyclic DSC (C_p) scan of $a_g(\text{IL@ZIF-8-LT})$ obtained by heating-cooling-heating with a same rate of $20\text{ }^\circ\text{C}\cdot\text{min}^{-1}$.

“For $a_g(\text{IL@ZIF-8-LT})$, shown in Supplementary Figure S5, the glass-liquid transition is not as clearly visible as in the HT-sample, which is because crystalline ZIF-8 (together with residual IL) remains the primary phase in this case, and neither ZIF-8 nor the IL exhibit a DSC feature in this temperature range (Figure 2c). However, in the second upscan, a weak glass transition is detected also for the LT sample, what is *in line* with our interpretation of progressive reactions during DSC scanning.”

Comment 3. *The authors choose this specific IL, [EMIM][TFSI], because it is hydrophobic. However, after the careful analysis using the solid-state NMR, this IL molecules effectively interact with the ZIF-8 framework via hydrogen bonds. Did the authors try another IL that is hydrophobic but not strongly interact with the framework? In this manuscript, only one IL was used so that it is more difficult to really understand the effect of IL on the melting behavior.*

Response. We appreciate reviewer's comment. To better understand the role of IL in melting of ZIF-8, we synthesized a new composite using [BMIM][PF₆] as a hydrophobic IL with the same loading of 35 wt%. Figure 1a shows that crystallinity of the ZIF-8 is preserved after [BMIM][PF₆] incorporation. FTIR of the [BMIM][PF₆]@ZIF-8 sample in Figure 1b shows the presence of [BMIM][PF₆] bands in the composite, confirming the successful incorporation of [BMIM][PF₆] into ZIF-8. To investigate the meltability of [BMIM][PF₆]@ZIF-8, we performed a DSC scan presented in Figure 1c. According to DSC scan, there is no melting transition, and the composite decomposes at around 350 °C (that is, before melting). Thus, melting is not possible for this composite, probably as a result the low thermal stability of the IL which does not allow the composite to reach the melting temperature.⁶

Figure 1. Structural and thermal characterization of [BMIM][PF₆]@ZIF-8 composite: (a) XRD patterns of ZIF-8 and [BMIM][PF₆]@ZIF-8 composite. (b) FTIR spectra of ZIF-8, [BMIM][PF₆], and [BMIM][PF₆]@ZIF-8 composite. (c) DSC-TGA analysis of [BMIM][PF₆]@ZIF-8 composite.

According to these results, meltability of MOFs/ZIFs not only depends on the similar hydrophilicity of the framework and the IL, but also on its thermal stability (and interactions at high temperatures); this confirms our discussion of NMR data in the manuscript. At this point, we did not include the additional [BMIM][PF₆]@ZIF-8 investigation into the manuscript because it might affect the paper's clarity. However, if the reviewer and editor feel that such additional information might be to the reader's benefit, we could include them in the supplement.

We included the following text in the Discussion part on page 24:

“We note that similar hydrophobicity and high thermal stability of IL are important factors determining the meltability of IL incorporated MOF composites.”

Comment 4. *In the Figure 5b, the data of the HT sample is missing. Was this crystalline peak disappeared in the HT sample?*

Response. We appreciate the reviewer’s comment. We conducted the missing ^1H - ^{15}N CP NMR of the $a_g(\text{IL@ZIF-8-HT})$ sample, and the result is now included as Supplementary Figure S26 in the Supplementary Information.

Supplementary Figure S26. ^1H - ^{15}N CP NMR of $a_g(\text{IL@ZIF-8-HT})$.

^1H - ^{15}N CP NMR of the $a_g(\text{IL@ZIF-8-HT})$ agrees well with ^1H - ^{13}C CP NMR of $a_g(\text{IL@ZIF-8-HT})$ representing broad amorphous bands and the free linker peak. We added the following text on page 14 of the manuscript:

“Comparing to ^1H - ^{15}N CP NMR of $a_g(\text{IL@ZIF-8-LT})$, the sharp peak which would be evidence for a crystalline phase is not detected in ^1H - ^{15}N CP NMR of $a_g(\text{IL@ZIF-8-HT})$, shown in Supplementary Figure S26, which is in agreement with XRD data in **Figure 2a**.”

Reviewer #3 (Remarks to the Author):

In this manuscript, Nozari, et al, incorporated organic ionic liquid into the pores of ZIF-8 which is a non-meltable MOF, expecting the formation of ZIF-8 liquid prior decomposition at elevated temperature and hence the formation of glass after quenching the molten. Attempts are then made to understand the melting and the glass formation processes of ZIF-8 loaded with ionic liquid using DSC, FTIR, NMR, PDF, and XRD measurements. In addition, they examined the porosity of the melt-quenched glass using N_2 gas adsorption measurement. The idea of this manuscript may be good which will help to convert the non-meltable MOF crystals into glasses. But, I have several significant issues with this manuscript. My concerns are as follows:

Comment 1.- *The glass transition (@ 322 °C, Fig. 2C) of the quenched glass $a_g(\text{IL@ZIF-8-HT})$ is not clear! C_p trace is necessary to determine the accurate T_g and whether the observed endothermic peak is correlated with the amorphous MOF structure or it is due to the experimental error. Otherwise, the manuscript will lose its purpose and the melting of IL@ZIF-8 will be useless. In addition, the XRD and C_p measurements are needed for the heated ionic liquid at 390 °C to ensure the heated ionic liquid doesn't have amorphous or glassy nature.*

Response. We thank the reviewer for this comment. This comment is essentially equivalent to comment 1 of reviewer 1. Please refer to our response to the first comment of reviewer 1. In short, we now added C_p cycling data, which unambiguously show that the observed T_g is not a misinterpretation or due to experimental error. We thank the reviewer for pointing this out. Furthermore, we performed XRD of $[\text{EMIM}][\text{TFSI}]$ and presented those data in Figure 2.

Figure 2. XRD pattern of [EMIM][TFSI].

As expected, the XRD pattern of [EMIM][TFSI] shows a fully amorphous pattern with absence of any crystalline phase. Moreover, a cycle C_p scan up to 360 °C (the temperature range which we conducted cycle C_p scans for $a_g(\text{IL@ZIF-8-HT})$ glass) with heating and cooling rate of 20 °C·min⁻¹ was performed for the bulk [EMIM][TFSI], presented in Figure 3.

Figure 3. Cycle C_p scan of [EMIM][TFSI] performed using heating and cooling rate of $20\text{ }^\circ\text{C}\cdot\text{min}^{-1}$ up to $360\text{ }^\circ\text{C}$.

There is no phase transition in the C_p scan of [EMIM][TFSI] in the temperature range of up to $360\text{ }^\circ\text{C}$, where the glass transition was investigated for the $a_g(\text{IL@ZIF-8-HT})$ glass. Melting and glass transition of [EMIM][TFSI], as a widely investigated IL, occurs at sub-zero temperatures; several studies reported T_m and T_g of $-17\text{ }^\circ\text{C}$ and $-92\text{ }^\circ\text{C}$ for [EMIM][TFSI], respectively.⁷⁻⁹ Our results indicate that the IL does not show any glass transition in the temperature range studied in this work and the glass transition observed in the DSC or cycle C_p scans is related to the glassy characteristic of the $a_g(\text{IL@ZIF-8-HT})$ sample. We do not think that the data shown in above Fig. 2-3 would benefit the reader at this point, therefore, we did not include them into the revised manuscript. If the reviewer or the editor feel that this should be done, we would be happy to add them as supplementary data files.

Comment 2- *Why did you measure the TGA trace of the $a_g(\text{IL@ZIF-8-HT})$ glass only up to $400\text{ }^\circ\text{C}$ (Fig. 2D)? consistency of curves is important.*

Response. We appreciate the reviewer’s comment. TGA scans presented in Figure 2d are moved to Supplementary Information as Supplementary Figure S2 and highlighted in yellow. To better represent the TGA scans and be consistent in terms of temperature, we separated the TGA scan of a_g(IL@ZIF-8-HT) glass as new panel b. Instead, we added a cycle C_p scan of a_g(IL@ZIF-8-HT) in Figure 2d. We note that, DSC-TGA was done on a_g(IL@ZIF-8-HT) to determine the T_g. That’s why we only performed the analysis until maximum temperature of 400 °C, where we expected T_g to occur. Otherwise, exceeding this temperature would more or less completely decompose the glass, which we are trying to avoid because it may irreversibly affect our C_p sample holder in the employed STA facility.

Comment 3- *The most significant concern is that—throughout the discussion/interpretation—I find consistently that the authors deeply focused on the melting mechanism of the IL@ZIF-8-HT, however, the investigation on the glassy nature of the composite sample is lacked. So, I recommend the authors to examine the influence of ionic liquid concentration on the glass formation process, transparency, and porosity (i.e. ZIF-8 loaded by 5, 10, 15, 20, 25, and 30 wt%).*

Response. We appreciate the reviewer’s comment. Loading of the IL in IL@ZIF-8 composite, 35 wt%, was selected as this was the maximum loading before reaching the insipient wetness point, as explained in the manuscript. To stabilize the interactions with the metal sites and organic linkers upon melting, it is essential to ensure the presence of sufficient IL molecules within the pores of ZIF-8. Based on the EMIMTFSI density and number of supercages in ZIF-8, 1×10^{23} supercages mol⁻¹,¹⁰ the corresponding number of IL molecules per supercage of different loadings of 5, 10, 15, 20, 25, 30, and 35 wt% is calculated and summarized in Table 1.

Table 1. Number of EMIMTFSI per supercage of ZIF-8 calculated based on different loadings.

IL loading (wt%)	IL per supercage of ZIF-8
35	1.89
30	1.50
25	1.16
20	0.87
15	0.61

10	0.38
5	0.18

Based on the number of ILs per supercage of ZIF-8, loadings lower than 25 wt% would probably fail to melt ZIF-8 because of the reduced average number of IL molecules per cage. Furthermore, during the synthesis, some of the IL molecules stay out of the pores and instead deposit on the external surface of ZIF-8. Thus, to guarantee the maximum presence of IL molecules inside the pores, the maximum amount of IL before reaching the wetness point, 35 wt %, was chosen. Nevertheless, we synthesized three more IL@ZIF-8 composites with different loadings of 5, 10, and 20 wt%, IL@ZIF-8-5 wt%, IL@ZIF-8-10 wt%, and IL@ZIF-8-20 wt% according to the synthesis procedure explained in the materials section of the manuscript. The obtained composites were characterized using XRD and DSC. Figure 4 shows XRD patterns of the synthesized composites.

Figure 4. XRD patterns of ZIF-8, IL@ZIF-8-5 wt%, IL@ZIF-8-10 wt%, and IL@ZIF-8-20 wt%.

Figure 5 illustrates DSC results obtained for IL@ZIF-8-5 wt%, IL@ZIF-8-10 wt%, and IL@ZIF-8-20 wt%.

Figure 5. DSC-TGA of IL@ZIF-8 composite with different IL loadings.

The DSC-TGA results of the IL@ZIF-8 composite with different loadings show no phase transition such as melting until deep into the decomposition range (~ 400 °C). Decomposition of the IL@ZIF-8 is shifted to lower temperatures as the IL loading increases. This is an expected shift as the IL is less thermally stable than pristine ZIF-8. Moreover, mass loss observed for different loadings shows the following trend for the respective IL loading in the composite; the more IL in the composite, the more mass loss observed upon decomposition. Looking at the DSC trace of IL@ZIF-8-20 wt%, an endothermic peak with onset and offset temperatures of 390 and 409 °C can be detected. At first glance this can be considered as the melting transition. However, corresponding mass loss values at T_{onset} , 3 %, and T_{offset} , 7 %, indicates that this endotherm is related to or at least overlapping with the decomposition of the sample. To confirm, IL@ZIF-8-20 wt% was heated to 410 °C with the same heating rate of 5 °C \cdot min $^{-1}$ and cooled to 50 °C with 20 °C

Figure 6. Optical image of IL@ZIF-8-20 wt% after heating to 410 °C and cooled to 50 °C. The sample diameter is around 6 mm.

$\cdot\text{min}^{-1}$. The sample image after the heat-treatment is shown in Figure 6. The optical image of the heat-treated sample of IL@ZIF-8-20 wt% shows that no melting transition happens upon heating to 410 °C.

Based on the DSC data of newly synthesized samples, lower IL loadings are not sufficient to induce melting of IL@ZIF-8. A specialized study of the exact effects of the amount of IL relative to ZIF-8 might be beneficial in the future, but we point-out that such a study would need to take into account many parameters aside the pure mixing ratio, *e.g.*, including the ZIF-8 crystallite size and morphology, the exact mixing parameters etc. We feel that this is outside of the scope of our present report on the discovery of IL-facilitated melting of ZIF-8.

Comment 4- *“It offers exciting opportunities to melt other non-meltable crystalline MOFs, enabling a broad range of new hybrid glasses” They didn’t clearly confirm whether the melt-quenched sample possesses glassy nature.*

Response. We appreciate the reviewer’s comment. This comment is essentially equivalent to comment 1 of reviewer 1, and also to the first comment of this reviewer. Please refer to our response to the first comment of reviewer 1. By definition, the glassy nature of a material is proven when

the occurrence of a glass transition is demonstrated. We hope that our newly added C_p scans, together may convince this reviewer in this regard. Further confirmation is from PDF analysis in Figure 8, showing the loss of any long-range order and retention of short-range order in $a_g(\text{IL@ZIF-8-HT})$ glass.

Comment 5- *What is the potential application of the ZIF-8 glass loaded with ionic liquid? What are the unique physicochemical properties of this MOF glass? The glass is almost opaque with very small pores volume ($0.001 \text{ cm}^3 \text{ g}^{-1}$) and it can't be prepared in large scale.*

Response. We appreciate the reviewer for this comment. Glasses produced by melting MOFs have emerged very recently and it is a nascent field of research. Finding a suitable candidate for a specific application requires expanding the number of meltable MOFs and broadening the range of available glass chemistries. In this work, our aim was to introduce a new route for melting non-meltable MOFs which might offer important opportunities in the future, and to expand on the fundamental understanding of the melting mechanism at the molecular scale. The search for applications of these new types of glasses is an emerging field. Nonetheless, we investigated the optical, mechanical, and microstructural properties of the $a_g(\text{IL@ZIF-8-HT})$ glass in order to provide incentives towards potential applications.^{5,11}

The following experimental details are included in Supplementary Information on page 4 and highlighted in yellow:

“Optical absorbance of $a_g(\text{IL@ZIF-8-HT})$ glass.

Optical properties of the $a_g(\text{IL@ZIF-8-HT})$ glass were analyzed by measuring the absorbance spectrum of $a_g(\text{IL@ZIF-8-HT})$ glass film on a platinum surface, shown in Supplementary Figure S35a, using a fiber-coupled spectrometer. We selected an area where the thickness is sufficient that the interference effects can be neglected. In Supplementary Figure S35a, interference colors on the edges of the sample can be seen due to multiple reflections between the platinum surface and the glass substrate. Laser scanning microscopy determined a sample thickness of around $6 \mu\text{m}$ at the point of measurement. Topography of the glass film and the thickness profile can be seen in Supplementary Figure S35b and S35c, respectively. Assuming that the reflectivity of platinum is approximately 1 and that the reflection and transmission losses on the glass surface is negligible, we obtain Absorbance $(\lambda) = \log [R(\lambda)] / (2d)$, where $R(\lambda) = \phi_{T(\lambda)} / \phi_{P(\lambda)}$ (see Supplementary Figure

S35d) which is the ratio between the flux reflected from an area covered with the glass and the flux reflected from an uncovered platinum surface. The measurement of the primary reflection could be suppressed by focusing through the glass on the glass-platinum interface. The light source was a deuterium arc lamp, covering a wide spectral range as shown in Supplementary Figure S35e. This allowed measuring reflection spectra from the sample in the range from 360 nm to 700 nm, as illustrated in Supplementary Figure S35f.’’

Results of the optical analysis are presented in Supplementary Figure S35 and highlighted in yellow as shown below:

Supplementary Figure S35. Optical absorbance of $a_g(\text{IL@ZIF-8-HT})$ film on a platinum surface using a deuterium arc lamp. **(a-c)** Thickness determination of $a_g(\text{IL@ZIF-8-HT})$ using a laser scanning microscope. **(d-f)** Absorbance spectra of $a_g(\text{IL@ZIF-8-HT})$ film on a platinum surface.

Details of nanoindentation experiments are summarized in Methods section and highlighted in yellow:

Nanoindentation tests. Instrumented indentation experiments were carried at room temperature using a nanoindenter (G200, KLA Co.) equipped with a Berkovich diamond tip (Synton-MDP Inc.). Samples were mounted in a thermosetting epoxy resin (Araldite CY212, Agar Scientific Ltd.). The tip area function and the instrument's frame compliance were calibrated prior to the first experiments on a fused silica reference glass (Corning Code 7980, Corning Inc.), following the method of Oliver and Pharr.¹² Depth profiles of hardness H and modulus E were obtained from indentations conducted in the continuous stiffness mode.¹³ In total, 25 indents with a depth limit of 2000 nm were performed per sample at a strain rate of 0.05 s^{-1} . Values of H were calculated from the load divided by the projected contact area of the indenter tip and values of E were derived from the reduced modulus.¹⁴

Depth profiles of H and E is included as Supplementary Figure S36 in the Supplementary Information. Based on the optical and mechanical characterization results, we included the following text in the Results section of the manuscript:

Supplementary Figure S36. Depth profiles of hardness (H) and modulus (E) for $a_g(\text{IL@ZIF-8-HT})$ glass obtained by instrumented indentation testing.

“We conducted further investigations on the physical properties of the $a_g(\text{IL@ZIF-8-HT})$ glass by examining optical absorbance as well as mechanical properties. Absorbance spectra taken of an $a_g(\text{IL@ZIF-8-HT})$ glass film on a platinum substrate are presented in Supplementary Figure S35 (experimental details are provided as Supplementary Information). The $a_g(\text{IL@ZIF-8-HT})$ glass is generally optically transparent, but absorbing in the blue to the green part of the visible spectrum. This results in the brownish-red coloration observed in the microscope images presented in **Figure**

3. Interestingly, the absorbance decreases towards the UV region, in the spectral range of ~ 500–360 nm (limit of the detector), which is a clear indicator for the absence of light scattering by crystal inclusions or other heterogeneities with a characteristic size corresponding to the wavelength of observation. Coloration aside, the obtained glass samples are highly transparent and possess no evidence of large-scale phase separation.

The mechanical properties of the $a_g(\text{IL@ZIF-8-HT})$ glass were studied through instrumented indentation testing using a nanoindenter (*see* Methods for details). The obtained depth profiles of hardness (H) and elastic modulus (E) are depicted in Supplementary Figure S36. The indentation response displays a marked load dependence, which reflects in a continuous decrease of H and E with increasing displacement into the glass' surface (h). This effect is more pronounced at small indentation depths and tends to level off as the indenter penetration increases, with the values of E and H eventually approaching the values of crystalline ZIF-8 reported in literature ($H = 0.501$ GPa and $E = 3.10$ GPa).¹⁵ Averaging the values of E and H between the upper 10 % and lower 20 % of each indentation depth profile resulted in mean values of $H = 0.730 \pm 0.136$ GPa and $E = 5.42 \pm 1.10$ GPa for $a_g(\text{IL@ZIF-8-HT})$. In a direct comparison with crystalline ZIF-8, we must emphasize that the mechanical stability may strongly vary as a function of the employed synthesis conditions: values of H ranging from 0.501 GPa to 0.643 GPa have been reported for crystalline ZIF-8. Likewise, E was reported with values of only 3.10 GPa up to about 7.33 GPa.^{15,16} In this regard, the present $a_g(\text{IL@ZIF-8-HT})$ falls in the same category as other ZIF-type glasses, such as $a_g\text{ZIF-4}$ ($H = 0.676$ GPa and $E = 7.17$ GPa) or $a_g\text{ZIF-62}$ ($H = 0.656$ GPa and $E = 6.85$ GPa).¹⁷

Regarding the porosity of $a_g(\text{IL@ZIF-8-LT})$ and $a_g(\text{IL@ZIF-8-HT})$ samples, please refer to our response to “Remarks to the Author” of reviewer 2.

Comment 5- *The authors state they obtained the MOFs glass. The glass samples they showed are grey or dark due to the decomposition. At least, glass samples should be transparent, and exhibit clearly glass transition phenomenon.*

Response. We thank the reviewer for this comment. Although the $a_g(\text{IL@ZIF-8-HT})$ glass reported in this work has a brownish color, measured optical properties illustrated in the response to the previous comment clearly showed optical transparency. In general, glasses can be colored by adding small concentrations of dopants.¹⁸ Furthermore, in the observed increase in optical

transparency when approaching the UV edge (~ 450-350 nm), we find evidence for the absence of crystal or other light-scattering inclusions. In response to the first comment, the glassy nature of a_g(IL@ZIF-8-HT) was investigated by performing cycle C_p scans, which unambiguously showed a glass transition. We hope that these new data may convince the reviewer that the reported material is indeed a glass.

Comment 6- *We do not see any promising applications of such glasses. Consequently, I do not recommend the acceptance of the paper for publication in the Nature Communications.*

Response. We respectfully disagree with this comment and have pointed towards initial suggested applications elsewhere in this response letter. However, we would also like to point out that i) the study is a fundamental one and fits within the remit of Nature Communication as an advancement in the field of MOF glasses and ii) when metallic glasses were first produced the reaction from some was very similar. As we have stated, melt-quenched MOF glasses have emerged very recently, and it is a nascent field of research. Even so, several promising applications have been presented in the literature, including gas separation and battery applications.^{5,11} We hope that the additional data we present in the revised version of this manuscript might further convince this reviewer of the benefits of our present approach.

References

1. Bennett, T. D. *et al.* Hybrid glasses from strong and fragile metal-organic framework liquids. *Nat. Commun.* **6**, 1–7 (2015).
2. Qiao, A. *et al.* A metal-organic framework with ultrahigh glass-forming ability. *Sci. Adv.* **4**, 1–8 (2018).
3. Longley, L. *et al.* Flux melting of metal-organic frameworks. *Chem. Sci.* **10**, 3592–3601 (2019).
4. Zhou, C. *et al.* Metal-organic framework glasses with permanent accessible porosity. *Nat. Commun.* **9**, 1–9 (2018).
5. Wang, Y. *et al.* A MOF Glass Membrane for Gas Separation. *Angew. Chem. Int. Ed.* **59**,

- 4365–4369 (2020).
6. Maton, C., De Vos, N. & Stevens, C. V. Ionic liquid thermal stabilities: Decomposition mechanisms and analysis tools. *Chem. Soc. Rev.* **42**, 5963–5977 (2013).
 7. Bhattacharjee, A., Lopes-da-Silva, J. A., Freire, M. G., Coutinho, J. A. P. & Carvalho, P. J. Thermophysical properties of phosphonium-based ionic liquids. *J. Chem. Eng. Data* **49**, 954–964 (2004).
 8. Béguin, F., Pavlenko, V., Przygocki, P., Pawlyta, M. & Ratajczak, P. Melting point depression of ionic liquids by their confinement in carbons of controlled mesoporosity. *Carbon N. Y.* **169**, 501–511 (2020).
 9. Monti, D., Jónsson, E., Palacín, M. R. & Johansson, P. Ionic liquid based electrolytes for sodium-ion batteries : Na⁺ solvation and ionic conductivity. *J. Power Sources* **245**, 630–636 (2014).
 10. Ban, Y. *et al.* Confinement of Ionic Liquids in Nanocages: Tailoring the Molecular Sieving Properties of ZIF-8 for Membrane-Based CO₂ Capture. *Angew. Chemie - Int. Ed.* **54**, 15483–15487 (2015).
 11. C, G. *et al.* Metal-Organic Framework Glass Anode with an Exceptional Cycling-Induced Capacity Enhancement for Lithium Ion Batteries. *ChemRxiv* (2021).
 12. Oliver, W. C. & Pharr, G. M. An improved technique for determining hardness and elastic modulus using load and displacement sensing indentation experiments. *J. Mater. Res.* **7**, 1564 (1992).
 13. Limbach, R., Rodrigues, B. P. & Wondraczek, L. Strain-rate sensitivity of glasses. *J. Non. Cryst. Solids* **404**, 124–134 (2014).
 14. Longley, L. *et al.* Metal-organic framework and inorganic glass composites. *Nat. Commun.* **11**, 1–12 (2020).
 15. Tan, J. C., Bennett, T. D. & Cheetham, A. K. Chemical structure, network topology, and porosity effects on the mechanical properties of zeolitic imidazolate frameworks. *Proc. Natl. Acad. Sci. U. S. A.* **107**, 9938–9943 (2010).

16. Tian, T., Velazquez-Garcia, J., Bennett, T. D. & Fairen-Jimenez, D. Mechanically and chemically robust ZIF-8 monoliths with high volumetric adsorption capacity. *J. Mater. Chem. A* **3**, 2999–3005 (2015).
17. Li, S. *et al.* Mechanical Properties and Processing Techniques of Bulk Metal-Organic Framework Glasses. *J. Am. Chem. Soc.* **141**, 1027–1034 (2019).
18. Möncke, D. Glass Color. *Encycl. Archaeol. Sci.* 1–3 (2018).

REVIEWER COMMENTS

Reviewer #1 (Remarks to the Author):

The authors have sufficiently addressed all prior concerns from this reviewer. As it stands, the manuscript is a major contribution to the field.

Reviewer #2 (Remarks to the Author):

The revised manuscript showed a lot of additional evidence to support the claim of the authors. In particular, I would appreciate the authors' effort to conduct the washing experiments and to show better porosity after washing. These experiments are really strong not only to support the claim of the authors' "IL guest infiltration decreased the melting point of ZIF-8" but to give a new approach to fabricate a "porous" MOF glass by the reversible guest accommodation/removal process. At this moment, these experiments would be sufficient to support the story of this manuscript. I would look forward to reading the further study by these authors regarding a new porous MOF glass in the future. I also appreciate the authors' effort to try another IL of [BMIB][PF6]. I do not think it is necessary to include this data. Without this trial, the story would flow better.

Overall, in this revision, the authors provided convincing experimental data, and therefore, I would support the acceptance of this manuscript as is.

Reviewer #3 (Remarks to the Author):

The authors have answered the questions basically.

Response to Reviewers 1-3: We greatly appreciate the final, supporting statements of the three reviewers as attached below. Their comments and suggestions have been of great help to us.

Reviewer #1 (Remarks to the Author):

The authors have sufficiently addressed all prior concerns from this reviewer. As it stands, the manuscript is a major contribution to the field.

Reviewer #2 (Remarks to the Author):

The revised manuscript showed a lot of additional evidence to support the claim of the authors. In particular, I would appreciate the authors' effort to conduct the washing experiments and to show better porosity after washing. These experiments are really strong not only to support the claim of the authors' "IL guest infiltration decreased the melting point of ZIF-8" but to give a new approach to fabricate a "porous" MOF glass by the reversible guest accommodation/removal process. At this moment, these experiments would be sufficient to support the story of this manuscript. I would look forward to reading the further study by these authors regarding a new porous MOF glass in the future.

I also appreciate the authors' effort to try another IL of [BMIB][PF6]. I do not think it is necessary to include this data. Without this trial, the story would flow better.

Overall, in this revision, the authors provided convincing experimental data, and therefore, I would support the acceptance of this manuscript as is.

Reviewer #3 (Remarks to the Author):

The authors have answered the questions basically.